# Variability of plasmid fitness effects contributes to plasmid persistence in bacterial communities

Aida Alonso-del Valle [1], Ricardo León-Sampedro [1,2], Jerónimo Rodríguez-Beltrán [1,2], Javier DelaFuente [1], Marta Hernández-García[1,3], Patricia Ruiz-Garbajosa[1,3], Rafael Cantón [1,3], Rafael Peña-Miller [4✉] & Alvaro San Millán [1,2,5✉]

Plasmid persistence in bacterial populations is strongly influenced by the fitness effects associated with plasmid carriage. However, plasmid fitness effects in wild-type bacterial hosts remain largely unexplored. In this study, we determined the fitness effects of the major antibiotic resistance plasmid pOXA-48_K8 in wild-type, ecologically compatible enterobacterial isolates from the human gut microbiota. Our results show that although pOXA-48_K8 produced an overall reduction in bacterial fitness, it produced small effects in most bacterial hosts, and even beneficial effects in several isolates. Moreover, genomic results showed a link between pOXA-48_K8 fitness effects and bacterial phylogeny, helping to explain plasmid epidemiology. Incorporating our fitness results into a simple population dynamics model revealed a new set of conditions for plasmid stability in bacterial communities, with plasmid persistence increasing with bacterial diversity and becoming less dependent on conjugation. These results help to explain the high prevalence of plasmids in the greatly diverse natural microbial communities.

[1] Servicio de Microbiología. Hospital Universitario Ramón y Cajal and Instituto Ramón y Cajal de Investigación Sanitaria, Madrid, Spain. [2] Centro de Investigación Biológica en Red. Epidemiología y Salud Pública, Instituto de Salud Carlos III, Madrid, Spain. [3] Red Española de Investigación en Patología Infecciosa. Instituto de Salud Carlos III, Madrid, Spain. [4] Center for Genomic Sciences, Universidad Nacional Autónoma de México, Cuernavaca, Mexico. [5] Centro Nacional de Biotecnología–CSIC, Madrid, Spain. ✉email: rafael.penamiller@gmail.com; alvsanmillan@gmail.com

Plasmids are extra-chromosomal mobile genetic elements able to transfer between bacteria through conjugation[1]. Plasmids carry accessory genes that help their hosts to adapt to a myriad of environments, and thus play a key role in bacterial ecology and evolution[2]. A key example of the importance of plasmids in bacterial evolution is their central role in the spread of antibiotic resistance mechanisms among clinical pathogens over recent decades[3,4]. Some of the most clinically relevant resistance genes, such those encoding carbapenemases (ß-lactamase enzymes able to degrade carbapenem antibiotics), are carried on conjugative plasmids that spread across high-risk bacterial clones[5,6].

Despite the abundance of plasmids in bacterial populations and the potential advantages associated with their acquisition, these genetic elements generally produce physiological alterations in their bacterial hosts that lead to a reduction in fitness[7–9]. These fitness costs make it difficult to explain how plasmids are maintained in bacterial populations over the long-term in the absence of selection for plasmid-encoded traits, a puzzle known as "the plasmid-paradox"[10]. Different solutions to this paradox have been proposed. For example, compensatory evolution contributes to plasmid persistence by alleviating the costs associated with plasmid carriage, and a high conjugation rate can promote the survival of plasmids as genetic parasites[11–18].

Over the past decades, many studies have investigated the existence conditions for plasmids in bacterial populations[14,18–23]. However, understanding of plasmid population biology is held in check by limitations of the model systems used for its study. First, most experimental reports of fitness costs have studied associations between plasmids and bacterial strains from different ecological origins[9,12–14,16,17,24–27] (with notable exceptions[15,28,29]). These examples do not necessarily replicate plasmid fitness effects in natural bacterial hosts, which remain largely unexplored. Second, studies tend to analyse the fitness effects of a single plasmid in a single bacterium. However, plasmid fitness effects can differ between bacteria[26,27,30–32], and this variability may impact plasmid persistence in bacterial communities (for a relevant example see ref. [33]). Third, most mathematical models of plasmid population biology study clonal or near-clonal populations. However, bacteria usually live in complex communities, in which conjugative plasmids can spread between different bacterial hosts[34–36]. To fully understand plasmid persistence in natural bacterial populations, it will be necessary to address these limitations.

In this study, we provide a detailed characterisation of the distribution of plasmid fitness effects in wild-type bacterial hosts. We use the clinically relevant carbapenem-resistance conjugative pOXA-48-like plasmid, pOXA-48_K8 (ref. [35]), and 50 enterobacteria strains isolated from the gut microbiota of patients admitted to a large tertiary hospital in Madrid. Incorporation of the experimentally determined plasmid fitness effects into a population biology model provides new key insights into the existence conditions of plasmids in bacterial communities.

## Results

**Construction of a pOXA-48_K8 transconjugant collection**. We studied the fitness effects of the plasmid pOXA-48_K8 in a collection of ecologically compatible (environmentally co-occurring) bacterial hosts. pOXA-48-like plasmids are enterobacterial, broad-host-range, conjugative plasmids, belonging to the IncL plasmid family, and are mainly associated with *Klebsiella pneumoniae* and *Escherichia coli*[37–39]. pOXA-48-like plasmids encode the carbapenemase OXA-48 and are distributed worldwide, making it one of the most clinically important carbapenemase-producing plasmids[6,38]. The gut microbiota of hospitalised patients is a frequent source of enterobacteria clones carrying pOXA-48-like plasmids[6]. In recent studies, we described the in-hospital epidemiology of pOXA-48-like plasmids in a large collection of extended-spectrum ß-lactamase (ESBL)- and carbapenemase-producing enterobacteria isolated from >9000 patients in our hospital over a period of 2 years (R-GNOSIS collection, see "Methods")[35,40–42]. pOXA-48-carrying enterobacteria were the most frequent carbapenemase-producing enterobacteria in the hospital, with 171 positive isolates, and they colonised 1.13% of the patients during the study period (105/9275 patients). In this study, we focused on plasmid pOXA-48_K8, which is the most common pOXA-48-like plasmid in our hospital[35] (Fig. 1a).

To study the fitness effects of pOXA-48_K8, we selected 50 isolates from the R-GNOSIS collection as bacterial hosts (Fig. 1b). Our criteria were to select (i) pOXA-48-free isolates recovered from patients with no record of colonisation by a pOXA-48-carrying enterobacteria, to try to avoid selecting clones in which compensatory evolution had already reduced plasmid-associated costs; (ii) isolates from the most frequent pOXA-48-carrying species, *K. pneumoniae* and *E. coli*; and (iii) strains isolated from patients located in wards, in which pOXA-48-carrying enterobacteria were commonly reported[35]. The underlying rationale was to select clones which were naive to pOXA-48_K8, but ecologically compatible with it (i.e., isolated from patients coinciding on wards with others who were colonised with pOXA-48-carrying clones). We selected 25 *K. pneumoniae* and 25 *E. coli* isolates that are representative of the R-GNOSIS study and cover the *K. pneumoniae* and *E. coli* phylogenetic diversity in the collection (see "Methods", Fig. 1c, and Supplementary Data 1). It is important to note that, because of the nature of the R-GNOSIS collection, the isolates used in this study produce ESBLs. However, ESBL-producing enterobacteria are widespread not only in hospitals but also in the community[43], and most pOXA-48-carrying enterobacteria isolated in our hospital also produce ESBLs[40].

pOXA-48_K8 was introduced into the collection of recipient strains by conjugation (see "Methods"), and the presence of the plasmid was confirmed by PCR and antibiotic susceptibility testing (Supplementary Data 2). The presence of the entire pOXA-48_K8 plasmid was confirmed by sequencing the complete genomes of the 50 transconjugant clones, which also revealed the genetic relatedness of the isolates (Fig. 1c). In line with previous studies[35,44], the sequencing results revealed that a subset of isolates initially identified as *K. pneumoniae* in fact belonged to the species *Klebsiella quasipneumoniae* ($n = 4$) and *Klebsiella variicola* ($n = 1$). These species are also hosts of pOXA-48-like plasmids in our hospital[35], and so were maintained in the study (Fig. 1b).

**Measuring pOXA-48_K8 fitness effects**. To measure pOXA-48_K8 fitness effects, we performed growth curves and competition assays for all the plasmid-carrying and plasmid-free clones in the collection. We first performed growth curves in pure cultures to calculate maximum growth rate ($\mu_{max}$) and maximum optical density ($OD_{max}$), which can be used to estimate the intrinsic population growth rate ($r$) and carrying capacity ($K$), respectively (Supplementary Figs. 1 and 2). We also measured the area under the growth curve (AUC), which integrates information about $r$ and $K$. To estimate plasmid-associated fitness effects, we compared these parameters between each plasmid-carrying and plasmid-free pair of isogenic isolates (Fig. 2a). The results showed that pOXA-48_K8 produced a moderate decrease in the parameters extracted from the growth curves, which was more evident in *Klebsiella* spp. isolates (Fig. 2a paired Wilcoxon signed-rank exact test: *E. coli*, AUC, $V = 114$, $P = 0.20$; $OD_{max}$, $V = 76$, $P = 0.020$; $\mu_{max}$, $V = 186$,

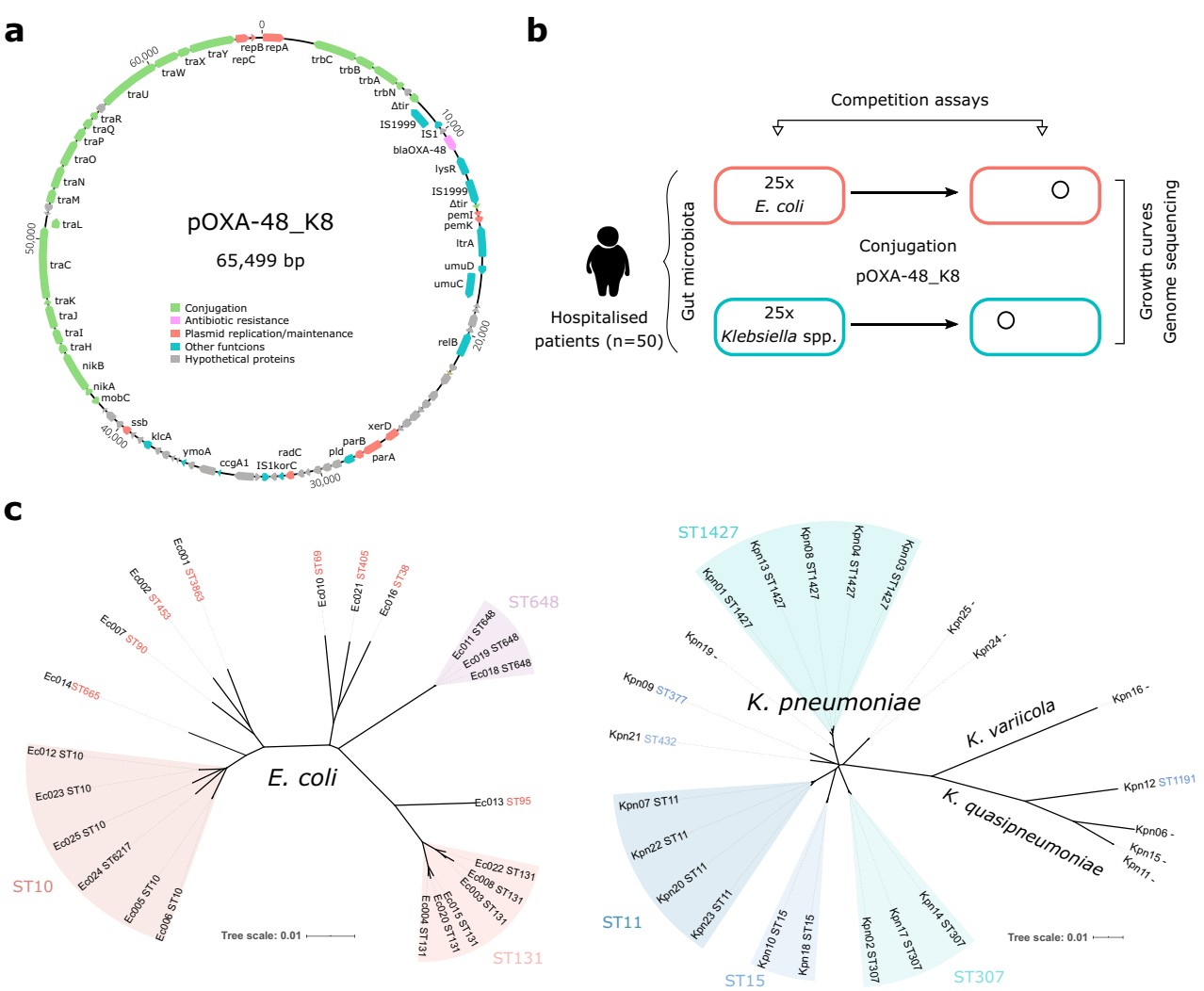

**Fig. 1 Experimental model system.** Representation of pOXA-48_K8 plasmid and the enterobacteria strains used in this study. **a** pOXA-48_K8 (accession number MT441554). Reading frames are shown as arrows, indicating the direction of transcription. Colours indicate gene function classification (see legend). The $bla_{OXA-48}$ gene is shown in pink. **b** Schematic representation of the experimental design used in this study. **c** Unrooted phylogeny of whole-genome assemblies from *E. coli* clones (left) and *Klebsiella* spp. clones (right). Branch length gives the inter-assembly mash distance (a measure of k-mer similarity). The grouping of multilocus sequence types (ST) is also indicated (*E. coli* ST6217 belongs to the ST10 group). Note that the sequencing results revealed that a subset of isolates initially identified as *K. pneumoniae* were in fact *Klebsiella quasipneumoniae* ($n = 4$) and *Klebsiella variicola* ($n = 1$).

$P = 0.54$. *Klebsiella* spp., AUC, $V = 54$, $P = 0.003$; $OD_{max}$, $V = 44$, $P = 0.001$; $\mu_{max}$, $V = 119$, $P = 0.25$). However, no differences were observed between the distribution of relative growth parameters for pOXA-48_K8-carrying *E. coli* and *Klebsiella* spp. clones (Wilcoxon rank-sum test: AUC, $W = 345$, $P = 0.38$; $OD_{max}$, $W = 335$, $P = 0.50$; $\mu_{max}$, $W = 347$, $P = 0.36$).

Competition assays allow measurement of the relative fitness ($w$) of two bacteria competing for resources in the same culture[45]. Competition between otherwise isogenic plasmid-carrying and plasmid-free clones thus provides a quantitative assessment of the fitness costs associated with plasmid carriage. For the competition assays, we used flow cytometry; strains were labelled using an in-house developed small, non-transferable plasmid vector, called pBGC, that encodes an inducible green fluorescent protein (GFP; Supplementary Fig. 3). pBGC was introduced into the wild-type isolate collection by electroporation, and all pOXA-48-carrying and pOXA-48-free clones were competed against their pBGC-carrying parental strain. We were unable to introduce pBGC into eight of the isolates; in those cases, for the competitor, we used *E. coli* strain J53 carrying the pBGC vector (see "Methods" for

details). Data from the competition assays were used to calculate the competitive fitness of pOXA-48-carrying clones relative to their plasmid-free counterparts (Fig. 2b). There were no significant differences between the fitness effects of pOXA-48_K8 in *Klebsiella* spp. and *E. coli* isolates (ANOVA effect of species × plasmid interaction; $F = 0.088$, $df = 1$, $P = 0.767$, Shapiro–Wilk normality test for $w$ values: *Klebsiella* spp., $P = 0.21$; *E. coli*, $P = 0.15$).

To validate our results, we compared the values obtained from growth curves and competition assays. This analysis revealed a weak but significant correlation between relative fitness values and the parameters extracted from the growth curves (Pearson's correlation, AUC, $R = 0.4$, $P = 0.005$; $OD_{max}$, $R = 0.33$, $P = 0.021$; $\mu_{max}$, $R = 0.4$, $P = 0.004$, Supplementary Fig. 4).

**The distribution of pOXA-48_K8 fitness effects.** Results from the competition assays showed that the overall effect of pOXA-48_K8 was a small, but significant reduction in relative fitness (mean $w = 0.971$, ANOVA effect of plasmid; $F = 70.04$, $df = 1$, $P = 1.02 \times 10^{-15}$). However, plasmid fitness effects varied greatly between the isolates in the collection, producing a normal

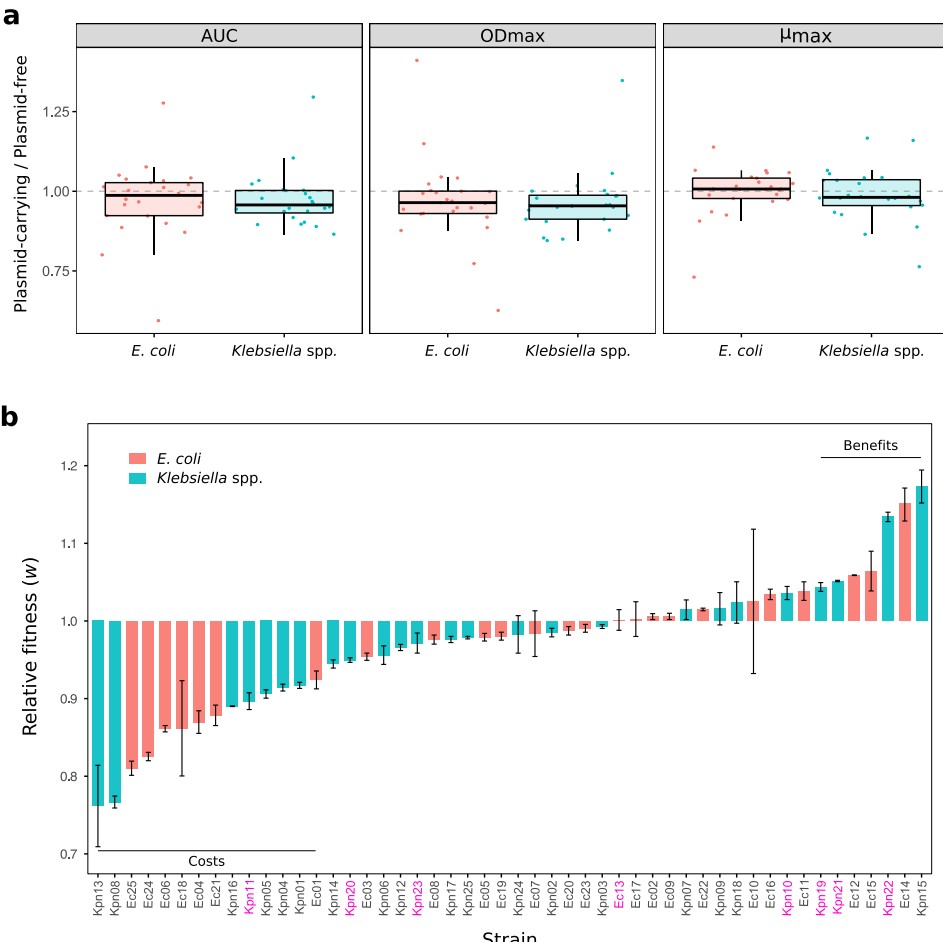

**Fig. 2 pOXA-48_K8 fitness effects in a set of ecologically compatible wild-type enterobacteria. a** Relative values of growth curve parameters (plasmid-carrying/plasmid-free isogenic clones): area under the growth curve (AUC), maximum optical density ($OD_{max}$), and maximum growth rate ($\mu_{max}$), represented as boxplots for *E. coli* (red) and *Klebsiella* spp. (blue) isolates separately. Horizontal lines inside boxes indicate median values, the upper and lower hinges correspond to the 25th and 75th percentiles, and whiskers extend to observations within 1.5 times the interquartile range. Dots represent each relative value. Values <1 indicate a reduction in these parameters associated with plasmid acquisition. Four and six biological replicates of the growth curves were performed for the wild-type isolates and the transconjugants, respectively (see Supplementary Figs. 1 and 2). **b** Relative fitness (*w*) of plasmid-carrying clones compared with plasmid-free clones obtained by competition assays (red, *E. coli*; blue, *Klebsiella* spp. see "Methods" for details). Values <1 indicate a reduction in *w* due to plasmid acquisition; values >1 indicate an increase in *w*. Bars represent normalised relative fitness after subtracting the effect of pBGC (average result of five independent biological replicates of the competition pOXA-48-carrying vs. pBGC-carrying strains divided by the average result of five independent biological replicates of the competition pOXA-48-free vs. pBGC-carrying strains), and error bars represent the propagated standard error. Two horizontal lines indicate those clones showing significant costs or benefits associated with carrying pOXA-48 plasmid (Bonferroni corrected two-sampled *t* test, $P < 0.05$). The names of the clones for which the relative fitness was calculated using *E. coli* strain J53 carrying the pBGC vector, instead of the pBGC-carrying parental strain, are indicated in pink. Source data are provided as Source data files.

distribution ranging from a >20% reduction to almost a 20% increase in relative fitness (Figs. 2b and 3a; Shapiro–Wilk normality test, $P = 0.14$). Indeed, plasmid acquisition was associated with a significant fitness decrease in only 14 strains, and 7 isolates showed a significant increase in fitness (Bonferroni corrected two-sampled *t* test, $P < 0.05$). These results revealed a highly dynamic scenario in which a plasmid produces a wide distribution of fitness effects in different bacterial hosts, ranging from costs to benefits.

To place our results in context with previous reports, we compared the distribution of pOXA-48_K8 fitness effects with the results from a recent meta-analysis of plasmid fitness effects by Vogwill and MacLean[46] (Fig. 3). These authors recovered data for 50 plasmid–bacterium pairs from 16 studies. The fitness effects obtained in those reports showed a higher mean plasmid cost

(mean $w = 0.91$) and differed significantly from the distribution of fitness effects we report here for pOXA-48_K8 in wild-type enterobacteria (Wilcoxon signed-rank test, $V = 922$, $P = 0.006$). The discrepancy between these distributions may, at least in part, reflect the different nature of plasmid–bacterium associations considered in the different studies. Although the plasmids studied in earlier reports were isolated from natural sources, they were introduced into bacterial hosts that are not necessary recovered from the same sources, and the detected fitness effects may not be fully representative of wild-type plasmid–bacterium associations. Our study, on the other hand, analysed the fitness effects of pOXA-48_K8 in ecologically compatible bacterial hosts. Taken together, the data suggest that the distribution of plasmid fitness effects is likely influenced by the ecological compatibility between plasmids and their bacterial hosts.

**a**

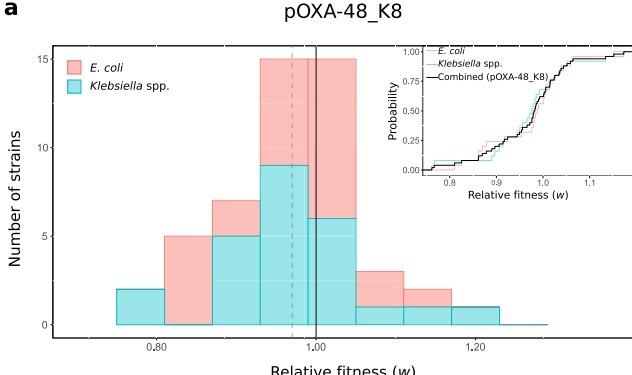

pOXA-48_K8

**b**

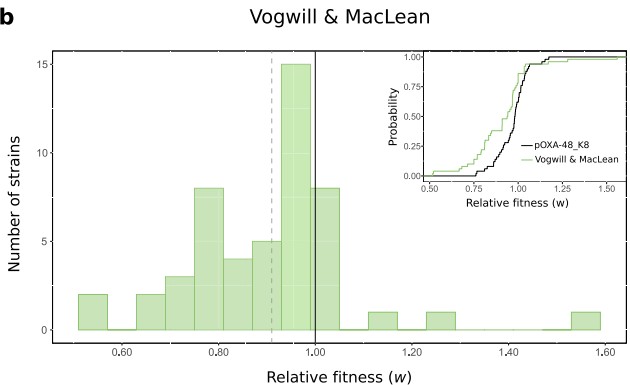

Vogwill & MacLean

**Fig. 3 Distribution of plasmid fitness effects.** Comparison between plasmid fitness effects obtained in this study and those from previous studies. **a** Distribution of pOXA-48_K8 fitness effects in the ecologically compatible collection of enterobacteria isolates. Bars indicate the number of *E. coli* (red) and *Klebsiella* spp. (blue) strains in each relative fitness category. The grey dotted line indicates the mean relative fitness of the population. Note that relative fitness values are normally distributed ($w =$ 0.971, var $= 0.0072$, one-sided Shapiro–Wilk normality test, $P = 0.14$). The inset shows the cumulative distribution function (CDF) of the relative fitness of pOXA-48_K8-carrying *E. coli* (red), *Klebsiella* spp. (blue) clones, both individually and combined (black). **b** Distribution of plasmids fitness effects in bacterial hosts obtained in a previous meta-analysis[46]. Most of the included studies were based on associations between plasmids and bacterial strains from different ecological origins. Bars indicate the number of plasmid–bacterium associations in each relative fitness category. The grey dotted line indicates the mean relative fitness across studies. Relative fitness values are not normally distributed ($w = 0.91$, var $= 0.029$; one-sided Shapiro–Wilk normality test, $P = 0.0006$). The inset shows the CDF of the relative fitness of pOXA-48_K8-carrying enterobacteria analysed in our study (black) and the CDF of the relative fitness of plasmid-carrying bacteria form Vogwill and MacLean meta-analysis (green). Source data are provided as a Source data file.

**pOXA-48_K8 fitness effects across bacterial phylogeny.** A key limit to the prediction of plasmid-mediated evolution is the inability to anticipate plasmid fitness effects in new bacterial hosts. This is particularly relevant to the evolution of antibiotic resistance because some of the most concerning multi-resistant clinical pathogens arise from very specific associations between resistance plasmids and high-risk bacterial clones[4,6,47]. Interestingly, a recent study in an important pathogenic *E. coli* lineage (ST131) showed that the acquisition and maintenance of resistance plasmids is associated with specific genetic signatures[48]. Pursuing this idea, we analysed pOXA-48_K8 fitness effects across the whole-genome phylogeny of our isolates, with the aim

of determining if genetic content could help to predict plasmid fitness effects (Fig. 4). We calculated the genetic relatedness of *Klebsiella* spp. and *E. coli* isolates by reconstructing their core genome phylogeny (Fig. 4a). Plasmid fitness effects can also be strongly influenced by the accessory genome. For example, the presence of further mobile genetic elements can deeply impact the costs of plasmids[24,49]. Therefore, we also constructed trees from the distance matrix of the accessory gene network[50], which includes plasmid content (Fig. 4b).

For each group of isolates, we scanned the fitness effects of pOXA-48_K8 across the core and accessory genome using the local indicator of phylogenetic association index[51,52] (LIPA; see Supplementary Fig. 5, Supplementary Data 3, and "Methods" for the complete analysis). For the *E. coli* isolates, the results showed no association of pOXA-48_K8 fitness effects with the core or accessory phylogenies (LIPA, $P > 0.1$). In contrast, for *Klebsiella* spp., LIPA indices revealed a significant phylogenetic signal in four clones, in which pOXA-48_K8 produced a high fitness cost, all of them belonging to ST1427 (Kpn01, Kpn04, Kpn08, and Kpn13, accounting for four of the five ST1427 clones analysed in this study; LIPA, $P < 0.05$). Three of these ST1427 clones also produced a significant signal in the analysis of fitness effects across the accessory genome (Kpn01, Kpn08, and Kpn13; LIPA, $P < 0.05$). Therefore, we analysed the specific association between the native plasmid content of each clone and pOXA-48_K8 fitness effects in *Klebsiella* spp. (Supplementary Fig. 6). Our analysis revealed that the presence of plasmids belonging to the IncFIA or IncH1B families, and the absence of plasmids belonging to the IncFIB family, were associated with high pOXA-48_K8 costs (Supplementary Fig. 6). However, since those specific plasmid profiles are closely associated with ST1427 clones in our collection, it is difficult to disentangle the relative contribution of core genome/accessory genome/plasmid content in determining the fitness costs of pOXA-48_K8 in this particular ST. Further work will be required to dissect the role of each of these genomic compartments.

The results revealed that pOXA-48_K8 tended to produce a high cost in *K. pneumoniae* clones belonging to ST1427. Interestingly, although *K. pneumoniae* ST1427 is relatively common in our hospital (4.8% of ESBL-producing *K. pneumoniae*[42]), none of the 103 pOXA-48-carrying *K. pneumoniae* isolates recovered in the R-GNOSIS collection belong to this ST[35] (Fisher's exact test for count data, 8/166 vs. 0/103, $P = 0.025$). These results suggest that the high cost associated with plasmid acquisition in this clade may limit in-hospital spread of pOXA-48-carrying *K. pneumoniae* ST1427. Conversely, pOXA-48-like plasmids are commonly associated with *K. pneumoniae* ST11 in our hospital[35,40], and in the four ST11 clones tested in this study, pOXA-48_K8 produced non-significant (Kpn07, Kpn20, and Kpn23) or even beneficial fitness effects (Kpn22, Fig. 4a; pOXA-48_K8 fitness effects in ST1427 [$n = 5$] vs. in ST11 [$n = 4$], Welch's unequal variances two-tailed $t$ test, $t = -2.39$, d$f = 7$, $P = 0.048$).

**Modelling the impact of variability of fitness effects on plasmid stability.** In general, mathematical models of plasmid population biology consider a clonal population in which the plasmid produces a constant reduction in growth rate[14,18–23]. These models usually include the rate of plasmid loss through segregation[53,54] and the rate of horizontal plasmid transfer by conjugation[19,20,55], and some of them also incorporate a rate of compensatory mutations that alleviate plasmid fitness costs over time[14,23]. Our results show that plasmids produce variable fitness effects in naturally compatible bacterial hosts, and we argued that this variability could strongly influence plasmid stability in natural

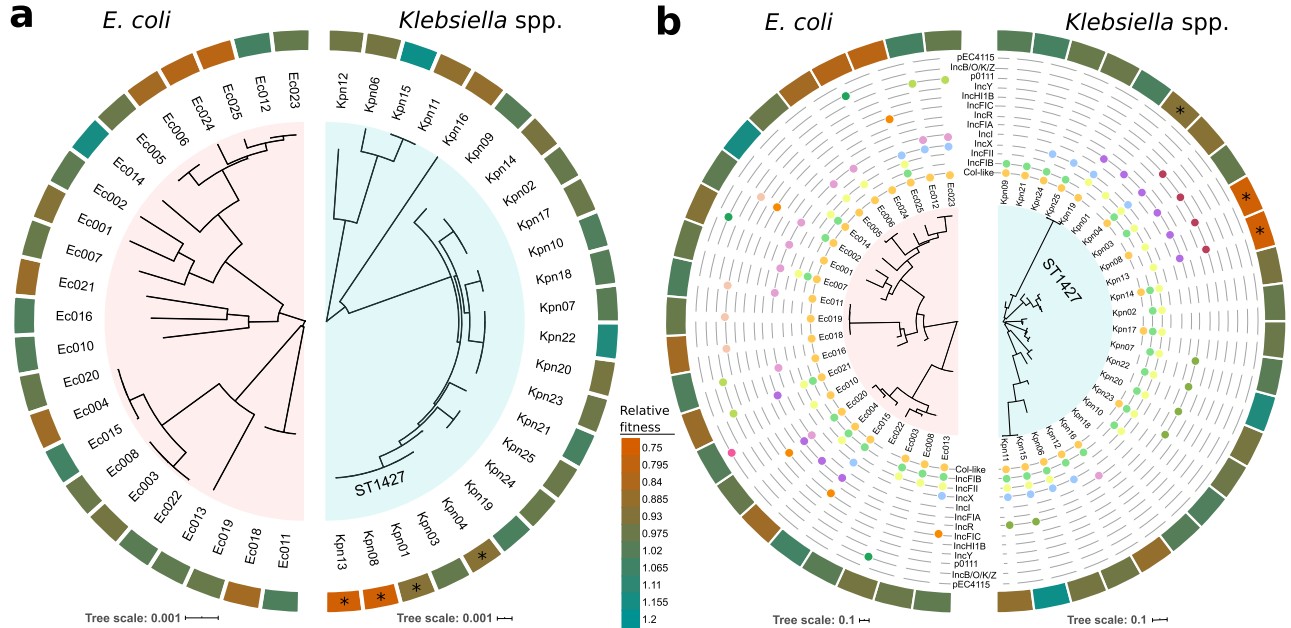

**Fig. 4 Fitness effects of pOXA-48_K8 across bacterial genome content.** An association was found between pOXA-48_K8 fitness effects and bacterial host genomic content for four *K. pneumoniae* ST1427 isolates. **a** Core genome relationships among *E. coli* (left) and *Klebsiella* spp. (right). Tree construction is based on polymorphisms in the core genome. The outer circle indicates the relative fitness of pOXA-48-carrying bacterial hosts (see legend for colour code; red indicates fitness costs and green indicates fitness benefits associated with pOXA-48_K8 carriage). Asterisks denote clones with a phylogenetic signal associated with plasmid fitness effects (LIPA, *P* < 0.05). **b** Accessory genome relationships among *E. coli* (left) and *Klebsiella* spp. (right) isolates. This tree is a gene content tree constructed based on the distance matrix of the accessory gene network of each group. The outermost circle indicates relative fitness as in **a**. The intermediate circles indicate presence/absence of plasmids belonging to the different plasmid families named in the figure. Note that only two isolates do not carry any plasmids. Asterisks denote clones with a significant phylogenetic signal associating accessory genome composition with pOXA-48 fitness effects (LIPA, *P* < 0.05).

polyclonal bacterial communities. To assess the effect of the variability of fitness effects on plasmid stability in bacterial communities, we developed a simple mathematical model based on Stewart and Levin's pioneering work on plasmid existence conditions[19]. Our specific aim was to inform this simple mathematical model with the experimentally determined distribution of pOXA-48_K8 fitness effects, in order to assess the consequences of the variability of fitness effects on the maintenance of plasmids in polyclonal bacterial communities (such as those in natural microbiota).

The model describes the population dynamics of multiple subpopulations competing for a single exhaustible resource in well-mixed environmental conditions, assuming that transition between plasmid-bearing and plasmid-free cells is driven by segregation events. The growth rate of each subpopulation is determined by a substrate-dependent Monod term that depends on the extracellular resource concentration, and therefore each strain can be described by two structurally identifiable parameters[53]: the resource conversion rate ($\rho$) and the specific affinity for the resource ($V_{max}/K_m$). These parameters were estimated from the ODs of each strain growing in monoculture (with and without plasmids) using a Markov chain Monte Carlo (MCMC) method with a Metropolis–Hastings sampler[56] (see "Methods", Fig. 5a, and Supplementary File 1).

By solving the system of differential equations (Eqs. (4–6), described in "Methods"), we were able to evaluate the final frequency of plasmid-bearing cells in an in silico competition experiment of duration *T* time units and quantify the fitness effect of the plasmid on the strain. Figure 5b shows the relative fitnesses obtained after performing computational pair-wise competition experiments between plasmid-bearing and plasmid-free subpopulations (with parameter values shown in Supplementary Table 1

and Supplementary Fig. 7), resulting in a theoretical distribution of fitness effects ($w = 0.985$, var = 0.0070) that is consistent with the experimentally measured one presented in Fig. 3 ($w = 0.971$, var = 0.0072; Supplementary Fig. 8, $R^2 = 0.603$).

Previous studies showed that the probability of plasmid fixation is correlated with the rate of horizontal transmission[19,20,53]. As previous models, we consider horizontal transmission of plasmids as a function of the densities of donor and recipient cells, with conjugation events occurring at a constant rate. Competition experiments between plasmid-free and plasmid-carrying clones for a range conjugation rates are illustrated in Fig. 5c; while at low horizontal transmission rates plasmid-free cells outcompete plasmid-bearing cells, at higher conjugative rates, plasmid-bearing cells increase in frequency. Crucially, allowing the fitness effects to vary (sampling for random plasmid–host associations from the MCMC parameter distribution) reduces the conjugation threshold that helps the plasmid to persist in the population, compared to assuming a fixed fitness effect (Fig. 5c). The effect of the variability of fitness effects on plasmid stability is particularly relevant for plasmids producing low average fitness cost (such as pOXA-48_K8), because the availability of hosts where the plasmid produces moderate effects or is beneficial is relatively high. For plasmids producing a large average fitness cost, the conjugation threshold required for plasmid maintenance is less affected by the variability of plasmid fitness effects in the different bacterial hosts (Fig. 5d and Supplementary Fig. 9).

**Community complexity promotes plasmid persistence.** Our results strongly suggest that community complexity should promote plasmid persistence by giving the plasmid access to a subpopulation of permissive bacterial hosts. To explore how plasmid

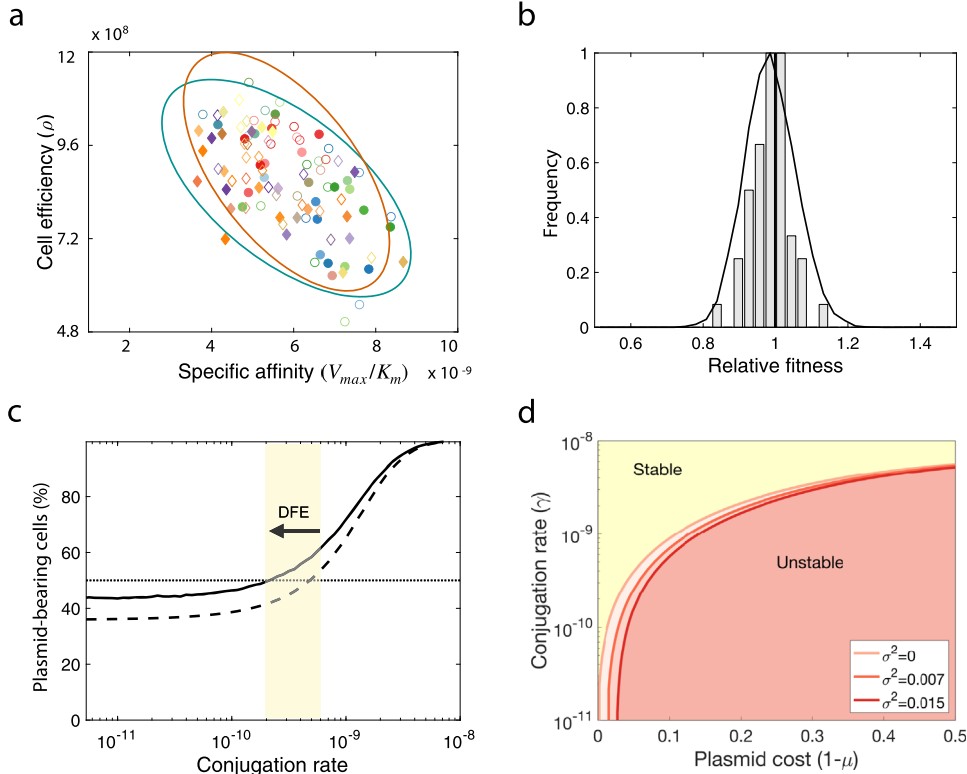

**Fig. 5 Modelling pOXA-48 fitness effects. a** Distribution of parameter values obtained using Bayesian inference to estimate growth kinetic parameters from OD measurements obtained for each strain in isolation. Diamonds represent *Klebsiella* spp. strains and circles *E. coli* clones; filled symbols denote plasmid-bearing strains and empty symbols plasmid-free cells. The ellipses represent standard deviations of best-fit normal distributions (green for plasmid-bearing strains and orange for plasmid-free cells). **b** Bars represent a distribution of plasmid fitness effects obtained from in silico competition experiments with parameter values determined from experimental growth curves. The solid curve represents the computationally estimated distribution obtained by randomly sampling wild-type and transconjugant parameter distributions obtained using the MCMC algorithm and numerically solving the model to evaluate the relative fitness associated with plasmid carriage. **c** Fraction of plasmid-bearing cells (after competing against plasmid-free cells) as a function of the rate of horizontal transfer for random plasmid–host associations sampled from the MCMC parameter distribution. The dotted line illustrates the mean of $10^4$ pair-wise competition experiments under the assumption that plasmid-bearing is associated with a constant reduction in fitness in different clones ($w = 0.985$, var $= 0$), while the solid line is obtained by considering a wide variability of fitness effects ($w = 0.985$, var $= 0.0070$). The arrow denotes the difference in the conjugation threshold that positively selects for plasmids in the population, supporting the tenet that the variability of fitness effects maintains plasmids in the population at lower conjugation rates. **d** Stability of plasmids as a function of plasmid cost and conjugation rate. Yellow area corresponds to the range of conjugation rates and plasmid costs that positively selects for the plasmid in the population, while red denotes the plasmid is unstable. Solid lines illustrate the critical conjugation rate estimated numerically after performing $10^3$ pair-wise competition experiments under the assumption that plasmid-bearing is associated with different levels of variability of fitness effects (see legend). Note that the increase in plasmid stability associated with the variability of fitness effects is more relevant for plasmids producing low average fitness cost, and becomes negligible for plasmids producing large average fitness cost, which depend more dramatically on a high conjugation rate. Source data are provided as a Source data file.

stability is affected by increasing community complexity and rates of horizontal transmission, we randomly sampled $N = 10^4$ plasmid-free cells from the distribution of growth parameters estimated using the MCMC algorithm. These random communities were used to study the population dynamics of plasmids transmitting vertically and horizontally in multi-strain communities. The fitness cost (or benefit) of bearing plasmids was modelled as a random variable that modifies the wild-type (plasmid-free) growth rate by a factor $\sigma$, such that if $\sigma = 0$, the distribution of fitness effects has zero variance (Fig. 6a), but if $\sigma > 0$, the result is a symmetrical heavy-tailed distribution with a right-hand tail expanding towards positive fitness effects (Fig. 6b), indicating the existence of plasmid–host associations in which plasmid carriage produces a fitness benefit.

To assess how the variability of fitness effects influences plasmid persistence in polymicrobial communities, we extended the model to consider populations composed of subsets of 1, 2, 3, 4,…, $M \leq N$ cell types sampled randomly from the wild-type parameter distribution (see "Methods", and Supplementary

Figs. 10 and 11). This enabled us to estimate the relative frequency of plasmid-bearing cells at the end of a long-term experiment and evaluate the stability of the plasmid in multi-strain communities with different population structures. Initial bacterial densities were determined by first running the system forward (with all strains initially present at equal densities and carrying pOXA-48) for $T = 24$ time units, and then clearing all plasmid-free cells from the population. This assumption is akin to patients receiving an antibiotic therapy that clears all plasmid-free cells from the bacterial community. The results obtained after 5000 computer simulations over a range of conjugation rates and numbers of cell types in the community are shown in Fig. 6. The simulations either assumed an identical fitness cost for all strains ($w = 0.985$, var $= 0$; Fig. 6a, c) or allowed plasmid fitness effects to vary according to the experimentally determined fitness effects ($w = 0.985$, var $= 0.0070$; Fig. 6b, d).

Although the mean fitness cost was the same in both conditions, the results of the computational experiments suggest

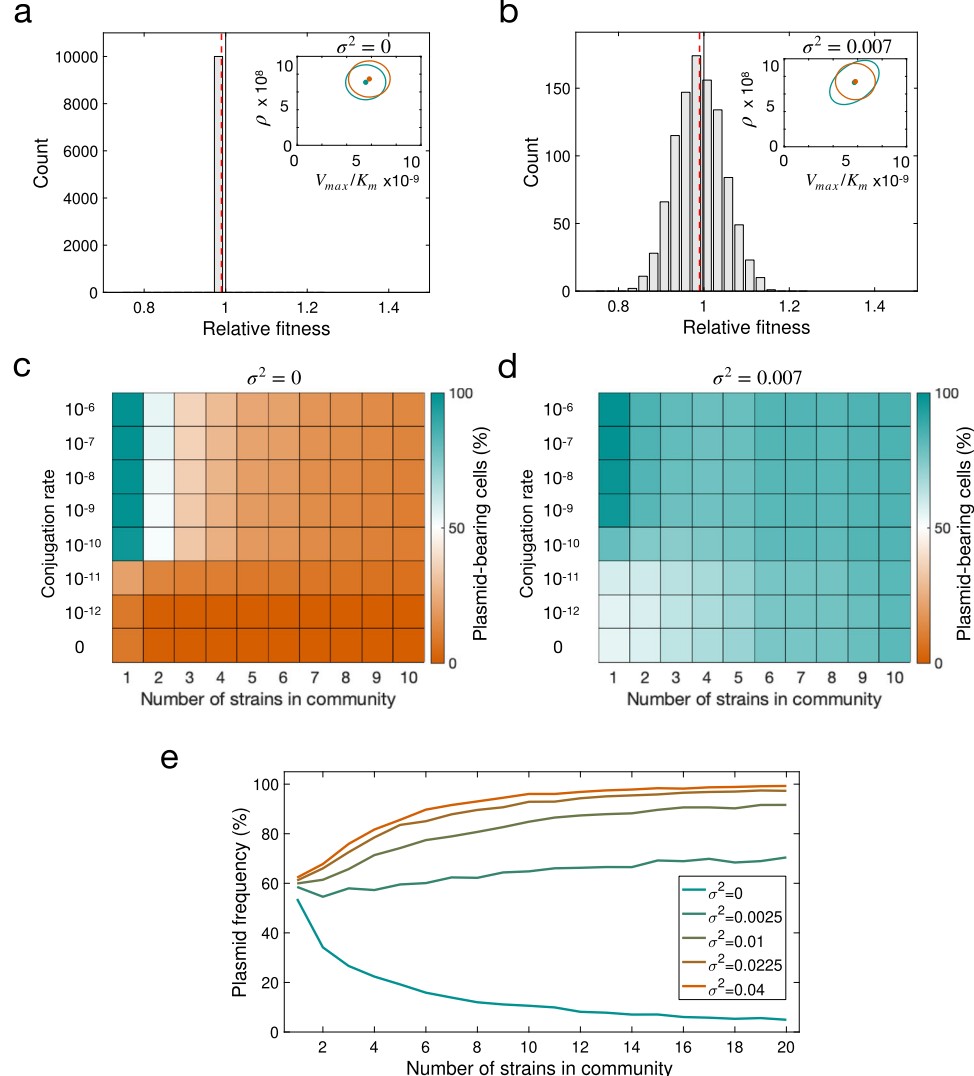

**Fig. 6 Plasmid persistence in complex communities.** Modelling plasmid persistence in polymicrobial communities, assuming fixed (**a**, **c**) or variable (**b**, **d**) plasmid fitness effects. **a**, **b** Relative fitness histogram obtained by randomly sampling $10^4$ parameter values from the parameter distribution shown in the inset plot (points illustrate the expected values of each distribution and ellipses their standard deviation; green, plasmid-bearing bacteria; orange, plasmid-free bacteria). The green ellipse in b is larger as a consequence of considering that the cost of plasmid-bearing is normally distributed with variance ($\sigma^2 = 0.007$). As a result, the distribution of plasmid fitness effects also has higher variance, with a considerable fraction of plasmid–host associations producing a benefit to the host. Dotted red lines indicate mean relative fitness of plasmid carrying cells. **c**, **d** Colour gradient represents the percentage of cells carrying plasmids at the end of 5000 stochastic simulations; orange indicates a population without plasmids and green a community composed of plasmid-carrying cells. If plasmid-bearing is associated with a fixed fitness cost for all members of the community, plasmid maintenance requires a high conjugation rate. The increased proportion of plasmid-bearing cells in **d** indicates that a distribution of plasmid fitness effects with high variance reduces the critical conjugation rate needed to maintain plasmids in the population, enabling plasmids to persist at low conjugation rates. **e** Mean fraction of plasmid-bearing cells as a function of the number of strains in the community with a conjugation rate $\gamma = 1.5 \times 10^{-11}$. If the plasmid always produces a reduction in host fitness (mean $w < 1$ and low variance), plasmid frequency decreases as the number of strains in the community increases (green line). In contrast, for higher variance at the same mean $w$, the fraction of plasmid-bearing cells increases with community complexity (orange line).

that allowing fitness effects to vary between members of the community markedly increases the chances of plasmid persistence, especially at low conjugation rates. More importantly, in the numerical simulations, plasmid frequency decreased as a function of the number strains in the community when plasmid acquisition was associated with a constant fitness cost, but increased with community complexity for distributions of fitness effects with larger variance (Fig. 6c–e). The explanation for this effect is that if plasmid fitness cost is identical for all community members, diversity simply means extra competition for plasmid-carrying cells, and plasmid persistence becomes more dependent on a high conjugation rate. In contrast, if the fitness effects vary, a larger number of available bacterial hosts in the population increases the probability of the plasmid arriving to a host, in which it produces no costs or beneficial fitness effect. This is an important result because it implies that increasing bacterial community complexity could increase the probability of plasmid persistence in natural environments. Given that most natural microbiota are complex and plasmids can usually conjugate and replicate in different clones, our results may help to explain the high prevalence of plasmids in nature. The results also indicate that the threshold conjugation rate for plasmid persistence may

be lower than previously thought. In fact, once plasmids are present in multiple members of a community, they may be able to persist even in the absence of conjugation (Fig. 6d).

## Discussion

In this study, we determined the fitness effects of a carbapenem-resistance plasmid in 50 wild-type enterobacteria recovered from the human gut microbiota. The distribution of pOXA-48_K8 fitness effects had a symmetrical shape, with a small average fitness cost and tails expanding towards negative and positive fitness effects (Fig. 3a). According to our simple mathematical model, two key implications arise from this experimentally determined distribution of plasmid fitness effects. First, the probability of plasmid persistence becomes less dependent on a high conjugation rate. This effect is particularly pronounced for plasmids producing a small average cost, but becomes negligible for plasmids producing a large cost (Fig. 5e). Second, and arguably most importantly, the probability of plasmid persistence increases with the number of bacterial strains in the community (Fig. 6). This result is in line with those from recent studies showing that community complexity may promote plasmid stability through source–sink plasmid transfer dynamics[29] and by providing multiple plasmid-permissive bacterial phylotypes[33,34].

The complex and multi-clonal nature of most natural bacterial communities attests the likely relevance of our findings to the extremely high prevalence of plasmids in bacterial populations[57]. The human gut microbiota, for example, includes a great variety of bacteria from hundreds of species[58], including several strains from the *Enterobacterales* order alone[59]. Our experimental system is in fact inspired by the dynamics of pOXA-48-like plasmids in the gut microbiota of hospitalised patients. In a recent study, we observed that once patients are colonised by a pOXA-48-carrying clone, the plasmid spreads through conjugation to other resident enterobacteria present in the gut microbiota[35]. Crucially, pOXA-48-like plasmids usually persist in the gut of patients throughout the hospital stay and can be detected in subsequent hospital admissions months or years later, and not necessarily in the original colonising strain[35]. Our results indicate that the variability of pOXA-48_K8 fitness effects could help to explain the long-term persistence of this and other plasmids in the human gut microbiota.

Another interesting result of this study is that pOXA-48_K8 produced a particularly elevated cost in *K. pneumoniae* isolates belonging to ST1427 (Fig. 4a). ST1427 is under-represented among the pOXA-48-carrying *K. pneumoniae* isolates in our hospital, which are dominated by ST11[35,40]. Remarkably, in the four *K. pneumoniae* ST11 clones tested in this study, pOXA-48_K8 produced non-significant (Kpn07, Kpn20, and Kpn23) or even beneficial fitness effects (Kpn22, Fig. 4a). Therefore, despite the small number of *K. pneumoniae* clones analysed, our results suggest that genetic differences between strains of the same species can explain differences in plasmid fitness cost, dictating the epidemiology of plasmid–bacterium associations in clinical settings. Further analysis of a larger sample of *K. pneumoniae* isolates from the different STs will be needed to elucidate the genetic basis underlying these specific interactions between bacterial phylogeny and pOXA-48_K8 fitness effects.

The main experimental limitation of our study is that plasmid fitness effects were determined in vitro, using planktonic cultures in lysogeny broth (LB) medium. This is the standard practise in the field, and previous studies have shown that plasmid fitness effects measured in laboratory conditions correlate with those measured in animal models[46]; however, our results may not be fully representative of pOXA-48_K8 fitness effects in the human gut. Future studies will need to explore more complex in vitro

systems[60], as well as in vivo animal models[61]. Another limitation is that, although in this study we used pOXA-48-free isolates recovered from patients with no record of colonisation by a pOXA-48-enterobacteria, it is impossible to completely rule out the possibility of the previous presence of a pOXA-48-like plasmid in these isolates, which could have led to pre-existing compensatory adaptations affecting the magnitude of pOXA-48_K8 fitness effects. Finally, another important limitation of our study is that we modelled bacterial communities with a simple resource competition model that does not consider spatial structure[62], complex ecological interactions between community members[63], plasmid–host co-evolution[64], or differential rates of horizontal transmission[31]. Although more complex models[65] will be needed to integrate bacterial community complexity and plasmid fitness effects, consideration of diverse polymicrobial populations with complex spatiotemporal interactions would likely only increase variance of fitness effects, therefore promoting plasmid stability.

## Methods

**Strains, pOXA-48_K8 plasmid, and culture conditions**. We selected 50 representative ESBL-producing clones form the R-GNOSIS collection (Supplementary Data 1). This collection was constructed in our hospital as part of an active surveillance-screening programme for detecting patients colonised by ESBL/carbapenemase-producing enterobacteria, from March 4th, 2014 to March 31st, 2016 (R-GNOSIS-FP7-HEALTH-F3-2011-282512, https://cordis.europa.eu/project/id/282512/reporting/es, approved by the Ramón y Cajal University Hospital Ethics Committee, Reference 251/13)[40,42]. The screening included a total of 28,089 samples from 9275 patients admitted at four different wards (gastroenterology, neurosurgery, pneumology, and urology) in the Ramon y Cajal University Hospital (Madrid, Spain). The characterisation of samples was performed during the R-GNOSIS study period[40,66]; rectal swabs were plated on Chromo ID-ESBL and Chrom-CARB/OXA-48 selective agar media (BioMérieux, France) and bacterial colonies able to grow on these media were identified by MALDI-TOF MS (Bruker Daltonics, Germany) and further characterised by pulsed-field gel electrophoresis (PFGE). For the present study, we selected 25 *E. coli* and 25 *K. pneumoniae* ESBL-producing isolates from the R-GNOSIS collection. The strains were representative of *E. coli* and *K. pneumoniae* diversity in the R-GNOSIS collection (randomly chosen form the most common PFGE profiles[42]), they did not carry any carbapenemase gene and they were recovered from patients not colonised by other pOXA-48-carrying clones. To construct the transconjugants, we used the most common pOXA-48-like plasmid variant from the R-GNOSIS collection in our hospital, according to plasmid genetic sequence (pOXA-48_K8, accession number MT441554)[26]. Bacterial strains were cultured in LB at 37 °C in 96-well plates with continuous shaking (250 r.p.m.) and on LB agar plates at 37 °C.

**Construction of transconjugants collection**. We performed an initial conjugation round to introduce pOXA-48_K8 plasmid from wild-type *E. coli* C609 strain[35], into *E. coli* β3914 (ref. [67]), a diaminopimelic acid (DAP) auxotrophic laboratory mutant of *E. coli* K-12 (kanamycin, erythromycin, and tetracycline resistant, Supplementary Data 1), which was used as the common counter-selectable donor. The pOXA-48-carrying wild-type *E. coli* C609 and *E. coli* β3914 were streaked from freezer stocks onto solid LB agar medium with ertapenem 0.5 µg/ml and DAP 0.3 mM, respectively, and incubated overnight at 37 °C. Donor and recipient colonies were independently inoculated in 2 ml of LB in 15-ml culture tubes and incubated overnight. After growth, donor and recipient cultures were collected by centrifugation (15 min, 1500 × g) and cells were re-suspended in each tube with 300 µl of sterile NaCl 0.9%. Then, the suspensions were mixed in a 1:1 proportion, spotted onto solid LB medium with DAP 0.3 mM and incubated at 37 °C overnight. Transconjugants were selected by streaking the conjugation mix on LB with ertapenem (0.5 µg/ml), DAP 0.3 mM, tetracycline (15 µg/ml), and kanamycin (30 µg/ml). The presence of pOXA-48_K8 was checked by PCR, using primers for *bla*$_{OXA-48}$ gene and for the replication initiation protein gene *repC* (Supplementary Table 2).

We used the counter-selectable *E. coli* β3914/pOXA-48_K8 donor to conjugate plasmid pOXA-48_K8 in the 50 wild-type strains. We used the same protocol described above, but the final conjugation mix was plated on LB with no DAP (to counter-select the donor) and with amoxicillin-clavulanic acid (to select for transconjugants). The optimal concentration of amoxicillin-clavulanic acid was experimentally determined for each isolate in the collection and ranged from 64 µg/ml to 384 µg/ml. The presence of pOXA-48_K8 in the transconjugants was checked by PCR, as described above, and by antibiotic susceptibility testing and whole-genome sequencing (see below). To test the stability of plasmid pOXA-48_K8 in the transconjugants, we propagated cultures in LB with no antibiotic selection (two consecutive days, 1:10,000 dilution) and plated cultures on LB agar. After ON incubation at 37 °C, 100 independent colonies of each transconjugant were replicated both on LB agar and LB agar with amoxicillin-clavulanic acid to identify

pOXA-48-carrying colonies (including negative controls of plasmid-free wild-type clones). Results showed that the plasmid was overall stable in the transconjugants; 100% stable in 43 isolates, and ≥90% stable in the 7 remaining isolates.

**Antibiotic susceptibility testing**. Antibiotic susceptibility profiles were determined for every wild-type and transconjugant strain by the disc diffusion method, following the EUCAST guidelines (www.eucast.org; Supplementary Data 2). We used the following antimicrobials agents: imipenem (10 μg), ertapenem (10 μg), amoxicillin-clavulanic acid (20/10 μg), rifampicin (30 μg), streptomycin (300 μg), chloramphenicol (30 μg), and amikacin (30 μg; BioRad, CA, USA). pOXA-48-carrying and pOXA-48-free strains were pre-cultured in Müller-Hinton (MH) broth at 37 °C in 15 ml test tubes with continuous shaking (250 r.p.m.), and disc diffusion antibiograms were performed on MH agar plates (BBL, Becton Dickinson, MD, USA).

**Growth curves**. Pre-cultures of plasmid-free and plasmid-carrying strains (four and six replicates of the plasmid-free and the plasmid-carrying strains, respectively) were prepared by inoculating single independent colonies into LB broth and overnight incubation at 37 °C with continuous shaking (250 r.p.m.). Overnight cultures were diluted 1:1000 into fresh LB in 96-well plates, which were incubated during 22 h at 37 °C with shaking (250 r.p.m.) in a plate reader (Synergy HTX Multi-Mode Reader, BioTek Instruments, Inc, VT, USA). ODs were measured every 15 min during the incubation. The maximum growth rate ($\mu_{max}$), maximum optical density ($OD_{max}$), and AUC were determined using the *growthrates* and *flux* packages in R. We calculated the relative $OD_{max}$, $\mu_{max}$, and AUC, by dividing the average value of each parameter for the pOXA-48-carrying isolate between that of the pOXA-48-free isolate using the following formula:

$$\text{Relative}_{OD\,max,Vmax,AUC} = \frac{\text{Plasmid} - \text{carrying}_{OD\,max,Vmax,AUC}}{\text{Plasmid} - \text{free}_{OD\,max,Vmax,AUC}} \quad (1)$$

**Construction of pBGC, a non-mobilisable, GFP-expressing plasmid**. To fluorescently label the wild-type isolates for competition assays using flow cytometry, we constructed the pBGC plasmid, a non-mobilizable version of the *gfp*-carrying small plasmid pBGT[25] (Supplementary Fig. 3, accession number MT702881). The pBGT backbone was amplified, except for the region including the *oriT* and *bla*$_{TEM1}$ gene, using the pBGC Fw/Rv primers. The *gfp* terminator region was independently amplified using the GFP-Term Fw/Rv primers (Supplementary Table 2). PCR amplifications were made with Phusion Hot Start II DNA Polymerase at 2 U/μl (ThermoFisher Scientific, MA, USA), and PCR products were digested with DpnI to eliminate plasmid template before setting up the assembly reaction (New England BioLabs, MA, USA). Finally, pBGC was constructed by joining the amplified pBGT backbone and the *gfp* terminator region using the Gibson Assembly Cloning Kit (New England BioLabs, MA, USA). Resulting reaction was transformed by heat shock into NEB 5-alpha Competent *E. coli* (New England BioLabs, MA, USA), following manufacturer's instructions. Transformation product was plated on LB agar with arabinose 0.1% and chloramphenicol 30 μg/ml, and incubated overnight at 37 °C. Plasmid-bearing colonies were selected by green fluorescence. The *gfp* gene in pBGC is under the control of the $P_{BAD}$ promoter, so GFP production is generally repressed and induced by the presence of arabinose. pBGC was completely sequenced using primers described in Supplementary Table 2. We confirmed that neither pOXA-48_K8, nor helper plasmid pTA-Mob[68], could mobilised pBGC by conjugation using the conjugation protocol described above, confirming that pBGC plasmid is not mobilizable. Finally, pBGC plasmid was introduced into our isolate collection by electroporation (Gene Pulser Xcell Electroporator, BioRad, CA, USA). Of note, we were not able to obtain pBGC-carrying transformants in eight of the isolates due, in part, to a pre-existing high chloramphenicol resistance phenotype (Ec13, Kpn10, Kpn11, and Kpn19–Kpn23).

**Competition assays using flow cytometry**. We performed competition assays[45], using flow cytometry, to obtain the relative fitness of pOXA-48-carrying isolates compared to their pOXA-48-free parental counterparts. We used the collection of pBGC transformed wild-type isolates as competitors against their isogenic pOXA-48-carrying and pOXA-48-free isolates. Specifically, two sets of competitions were performed for each isolate: pOXA-48-free vs. pBGC-carrying, and pOXA-48-carrying vs. pBGC-carrying (the "pBGC-carrying" are the wild-type pOXA-48-free isolates carrying pBGC). Five biological replicates of each competition were performed. Pre-cultures were incubated overnight in LB in 96-well plates at 225 r.p.m. and 37 °C, then mixed 1:1 and diluted 10,000-fold in 200 μl of fresh LB in in 96-well plates. Mixtures were competed for 24 h in LB at 37 °C and 250 r.p.m. (the low initial cell density and the strong shaking hinders pOXA-48_K8 conjugation, see control experiment below). To determine the initial proportions, initial 1:1 mix were diluted 2000-fold in 200 μl of NaCl 0.9 % with L-arabinose 0.1%, and incubated at 37 °C at 250 r.p.m. during 1.5 h to induce GFP expression (inducible GFP expression allows avoiding the costs associated with GFP production during the overnight competition). The measurements were performed via flow cytometry using a CytoFLEX Platform (Beckman Coulter Life Sciences, IN, US) with the following parameters: 50 μl/min flow rate, 22 μm core size, and 10,000 events recorded per sample (Supplementary Fig. 12). After 24 h of incubation, final

proportions were determined as described above, after 2000-fold dilution of the cultures. The fitness of each strain relative to its pBGC-carrying parental isolate was determined using the formula:

$$w = \frac{\ln(N_f/N_i)}{\ln(N_{f,pBGC+}/N_{i,pBGC+})} \quad (2)$$

where $w$ is the relative fitness of the pOXA-48-carrying ($w_{pOXA-48+}$) or pOXA-48-free ($w_{pOXA-48-}$) isolates compared to the pBGC-bearing parental clone, $N_i$ and $N_f$ are the number of cells of the pBGC-free clone at the beginning and end of the competition, and $N_{i,pBGC}$ and $N_{f,pBGC}$ are the number of cells of the pBGC-carrying clone at the beginning and end of the competition, respectively. The fitness of the pOXA-48-carrying isolates relative to the pOXA-48-free parental isolates were calculated with the formula, $w_{pOXA-48+}/w_{pOXA-48-}$ to correct for the fitness effects of pBCG (see Supplementary Fig. 13 for pBGC fitness effects). Specifically, the average result of five independent biological replicates of the competition pOXA-48-carrying vs. pBGC-carrying ($w_{pOXA-48+}$) was divided by the average result of five independent biological replicates of the competition pOXA-48-free vs. pBGC-carrying ($w_{pOXA-48-}$), and the error propagation method was used to calculate the standard error of the resulting value. Note that the fitness effects of pBGC did not correlate with those form pOXA-48 (Pearson's correlation, $R = 0.11$, $t = 0.66$, $df = 39$, $P = 0.51$). For the eight strains where pBGC plasmid could not be introduced (Ec13, Kpn10, Kpn11, and Kpn19–Kpn23), pOXA-48-carrying and pOXA-48-free isolates were competed against a pBGC-carrying *E. coli* J53 (ref. [69]; a sodium azide resistant laboratory mutant of *E. coli* K-12), following the same protocol described above. In general, we prefer to perform competitions assays between isogenic bacteria to avoid interactions between clones that may affect the outcome of the competition for reasons beyond the presence of the plasmid under study (such as bacteriocin production). However, we did not observe any evidence of growth inhibition between the eight wild-type isolates and *E. coli* J53 in the flow cytometry data, and the relative fitness results obtained with these competitions were comparable to those obtained in the isogenic competitions (two-tailed *t* test, $t = 1.64$, $df = 11.2$, $P = 0.13$). To confirm that the isogenic competitions and those against *E. coli* J53/pBGC produced similar results, we selected 10 random isolates from the 42 isolates with fitness data calculated from isogenic competitions, and repeated their competitions against *E. coli* J53/pBGC (Supplementary Fig. 14). Results showed that relative fitness values calculated with isogenic competitions and those using *E. coli* J53/pBGC presented a good correlation (Pearson's correlation, $R = 0.81$, $t = 3.96$, $df = 8$, $P = 0.004$, Supplementary Fig. 14). Finally, we performed controls to test for the potential conjugative transfer of pOXA-48_K8 during head-to-head competitions by plating the final time points of the competition assays on amoxicillin-clavulanic acid (with the adequate concentration for each isolate), and chloramphenicol (30 μg/ml). No transconjugants were detected in these controls, showing that the low initial inoculum size we used in the competitions (10,000-fold dilution), and the vigorous shaking of the liquid cultures prevented pOXA-48_K8 conjugation.

**DNA extraction and genome sequencing**. Genomic DNA of all the pOXA-48_K8-bearing strains was isolated using the Wizard genomic DNA purification kit (Promega, WI, USA), and quantified using the QuantiFluor dsDNA system (Promega, WI, USA), following manufacturers' instructions. Whole-genome sequencing was conducted at the Wellcome Trust Centre for Human Genetics (Oxford, UK), using the Illumina HiSeq4000 platform with 125 base pair (bp) paired-end reads and at MicrobesNG (Birmingham, UK), using Illumina platforms (MiSeq or HiSeq2500) with 250 bp paired-end reads.

**Bioinformatic analyses**. The Illumina sequence reads were trimmed using the Trimmomatic v0.33 tool[70]. SPAdes v3.9.0 (ref. [71]) was used to generate de novo assemblies from the trimmed sequence reads with the –cov-cutoff flag set to "auto". QUAST v4.6.0 (ref. [72]) was used to generate assembly statistics. Three genomes were dropped from the analysis because of the poor quality of the sequences (two *E. coli* [Ec09 and Ec17] and one *K. pneumoniae* [Kpn05]). All the de novo assemblies used reached enough quality, including total size of 5–7 Mb, and the total number of contigs over 1 kb was <200. Prokka v1.5 (ref. [73]) was used to annotate the de novo assemblies with predicted genes. The reads and the annotated genomes have been uploaded the National Center for Biotechnology Information (NCBI) under the BioProject ID PRJNA641166. The seven-gene ST of all the isolates was determined using the multilocus sequence-typing tool (https://github.com/tseemann/mlst). The plasmid content of each genome was characterised using PlasmidFinder 2.1 (ref. [74]), and the antibiotic resistance gene content was characterised with ResFinder 3.2 (ref. [75]) (Supplementary Data 1).

In order to confirm the presence of the entire pOXA-48_K8 plasmid, the sequences belonging to pOXA-48_K8 plasmid in the transconjugants were mapped using as reference the complete sequence of plasmid from the donor strain, which had been previously sequenced by PacBio[35] (from *K. pneumoniae* k8—GenBank accession number MT441554). Snippy v3.1 (https://github.com/tseemann/snippy) was used to check that no SNPs or indels accumulated in pOXA-48_K8 during strain construction. Coding sequences in pOXA-48_K8 were predicted and annotated using Prokka 1.14.6 software[73]. Plasmid annotation was complemented with the NCBI Prokaryotic Genome Annotation Pipeline[76].

To determine distances between genomes we used Mash v2.0 (ref. [77]) with the raw sequence reads, and a phylogeny was constructed with mashtree v0.33 (ref. [78]). For the analysis of the core genome we calculated the genetic relatedness of isolates belonging to *Klebsiella* spp. and to *E. coli* by reconstructing their core genome phylogeny with an alignment of the SNPs obtained with Snippy v3.1 (https://github.com/tseemann/snippy). A maximum likelihood tree was generated using IQ-TREE with automated detection of the best evolutionary model[79]. The tree was represented with midpoint root using the *phylotools* package in R (https://github.com/helixcn/phylotools) and visualised using the iTOL tool[80]. We also constructed a distance matrix of the accessory gene network to analyse the accessory genome. To this end, we used AccNET, a tool that allows to infer the accessory genome from the proteomes and cluster them based on protein similarity[50]. The set of representative proteins was used to build a binary matrix (presence/absence of proteins in the accessory genome) in the R environment and a cladogram to classify the strains according to the accessory genomes. The Euclidean distance was calculated by the "dist" function and a hierarchical clustering was performed with UPGMA using the "hclust" function in the R environment. This cladogram was represented with midpoint root using the *phylotools* package in R (https://github.com/helixcn/phylotools) and visualised using the iTOL tool[80].

**Analysis of plasmid fitness effects across bacterial phylogeny.** We tested for the presence of phylogenetic signal in core and accessory genomes of *E. coli* and *K. pneumoniae* using several statistical tests available in the *phylosignal* R package[51]. In essence, these analyses are designed to identify statistical dependence between a given continuous trait (relative fitness) and the phylogenetic tree of the taxa from which the trait is measured. Therefore, a positive phylogenetic signal indicates that there is a tendency for related taxa to resemble each other[81]. Several indices have been proposed to identify phylogenetic signal, but the choice among them is not straightforward[82]. We first assayed the methods implemented in the *phyloSignal* function, which produce global measures of phylogenetic signal (i.e., across the whole phylogeny). The methods employed were Abouheif's $C_{mean}$, Moran's $I$ index, Bloomberg's $K$ and $K^*$, and Pagel's $\lambda$ (ref. [51]). All methods except Pagel's $\lambda$ detected a marginally significant phylogenetic signal in the *K. pneumoniae* core genome (Supplementary Data 3 [first tab]; $0.11 > P > 0.02$). Abouheif's $C_{mean}$ and Moran's $I$ (but not Bloomberg's $K$ and $K^*$, and Pagel's $\lambda$) also detected a marginally significant signal in the *K. pneumoniae* accessory genome tree (Supplementary Data 3 [first tab]; $P < 0.056$ for both cases). Intrigued by these results, we used the LIPA based on local Moran's $I$, which is meant to detect local hotspots of phylogenetic signal[51,52]. LIPA, implemented in the *lipaMoran* function, computes local Moran's $I$ indexes for each tip of the phylogeny and a non-parametric test to ascertain statistical significance (Supplementary Fig. 5 and Supplementary Data 3 [second tab]).

To investigate the relationship between pOXA-48_K8 fitness effect and the native plasmid content we performed a factor analysis of mixed data (FAMD) using the *FactoMine* R package[83]. This method is similar to a multiple factor analysis or a classical principal component analysis, but taking into account and combining different types of variables (quantitative and qualitative). The graphical outputs including variables, factor map, and individuals factor map show the distribution of the bacterial isolates across the different factors, including the presence/absence of the different plasmid variants (qualitative variables), the relative fitness of plasmid-carrying isolates, and the total number of plasmids carried by each isolate (quantitative variables). The extraction and visualisation of the FAMD output was carried out with the *Factoextra* package on R environment (http://www.r-project.org).

**Plasmid population dynamics model.** We used a simple mathematical model of microbial growth under resource limitation to study the role of the DFE in the ecological dynamics of a plasmid spreading in a bacterial population[14]. Bacterial growth rate was modelled as a saturating function of the environmental resource concentration, $R$,

$$G(R) = \rho \cdot \frac{V_{max}R}{K_m + R} = \rho \cdot u(R), \qquad (3)$$

where $\rho$ denotes the cell's efficiency to convert resource molecules into biomass and $u(R)$ a resource uptake function that depends on the maximum uptake rate ($V_{max}$) and a half-saturation constant ($K_m$). If we denote with $B_p$ the density of plasmid-bearing cells and with $B_0$ the density of plasmid-free cells (each with its own growth kinetic parameters and growth functions denoted $G_p(R)$ and $G_0(R)$, respectively), then the density of each subpopulation can be described by a system of ordinary differential equations:

$$\frac{dR}{dt} = -u_p(R) - u_0(R) - dR, \qquad (4)$$

$$\frac{dB_p}{dt} = (1-\lambda)G_p(R)B_p + \gamma B_0 B_p - dB_p, \qquad (5)$$

$$\frac{dB_0}{dt} = G_0(R)B_0 + \lambda G_p(R)B_p - \gamma B_0 B_p - dB_0. \qquad (6)$$

where $\lambda$ represents the rate of segregational loss rate and $d$ a dilution parameter.

Moreover, we represent with $\gamma$ the rate of conjugative transfer, and therefore we model plasmid conjugation as a function of the densities of donor and recipient cells. By numerically solving the system of equations (using standard differential equations solvers in Matlab), we obtain the final density of each bacterial type in an experiment of $T = 24$ units of time with $d = 0$ (to replicate the batch culture conditions used to estimate the DFE experimentally).

**Model parametrization.** Parameters of the population dynamics model were jointly determined using a Metropolis–Hastings MCMC (scripts coded in R and available in a public repository[56]) by fitting a simple Monod model described by the cell's specific affinity ($V_{max}/K_m$) and its resource conversion rate ($\rho$) to growth curves of each strain growing in isolation, with and without plasmids. This data-fitting algorithm implements a Metropolis–Hastings sampler that was executed for $1 \times 10^7$ iterations, with a burn-in parameter of 0.2 and a thinning of 100 iterations. We used an adaptive proposal variance method as a first step of the MCMC algorithm to adjust the acceptance probabilities for the proposed updates. During this period no samples were stored in the chain until the acceptance ratio reached a target value, then we fixed the proposal variance and started storing the MCMC iterations. Figures in Supplementary File 1 display posterior distributions obtained for each parameter and Supplementary Table 1 the maximum likelihood values estimated for each strain. As illustrated in panel a (Supplementary File 1), traces of chains presented good mixing, with convergence of the Markov chains verified by obtaining Gelman–Rubin Rhat statistical values near one for all strains. We tested robustness to different priors by considering uniform, lognormal, beta, and gamma distributions, and obtaining similar maximum likelihood estimates for all prior distributions (Supplementary File 1c, d; ANOVA $P > 0.05$, $H_0$: there are significant differences in estimates when considering different priors). Identifiability of parameters was confirmed using a data cloning algorithm[84,85] such that, as the number of clones increases, the marginal posterior distribution converges to the maximum likelihood estimate (Supplementary File 1f, g; ANOVA $P > 0.05$ for all strains, $H_0$: there are significant differences in estimates when considering different number of clones). Data and scripts necessary to produce the MCMC diagnostic plots can be downloaded from http://www.github.com/ccg-esb/pOXA48/.

**Stochastic simulations of polymicrobial communities.** Numerical experiments were performed by randomly sampling $N = 1 \times 10^4$ cells from the parameter distribution obtained after applying the MCMC algorithm to all 50 strains and fitting a bivariate Normal distribution. We then assembled 5000 synthetic communities composed of a random subset of $M < N$ different strains sampled from this distribution, and solved a multi-strain extension of the population dynamics model. For each numerical experiment, the total density of strain $i$ would be $B^i(t) = B_p^i(t) + B_0^i(t)$, where $B_p^i$ and $B_0^i$ denote, respectively, the densities of plasmid-bearing and plasmid-free cells of type $1 \le i \le M$. To model the fitness effects of bearing plasmids, we introduced a parameter, $\sigma$, such that when $\sigma = 0$, the fitness difference between $B_p^i$ and $B_0^i$ corresponds to a fixed reduction in growth rate (corresponding to a DFE with variance 0 and mean $w = 0.985$). Conversely, if $\sigma > 0$, then growth kinetic parameters for each plasmid-bearing strain in the community were determined by sampling $s_i$ from a Normal distribution, $N(0, \sigma^2)$, and multiplying both $V_{max}^i$ and $\rho^i$ by a factor of $(1 + s_i)$. Supplementary Fig. 15 shows a sample of parameter values for plasmid-bearing and plasmid-free cells obtained from considering different mean and variances in a theoretical distribution of plasmid fitness effects.

As with the single-strain model, we consider segregational loss as a transition from $B_p^i$ to $B_0^i$ occurring at a rate $\lambda$, but now we also consider that plasmid-free cells can acquire plasmids via conjugation from any plasmid-bearing strain in the community, at a constant rate $\gamma$, and with equal probability of transferring between different bacterial hosts. Therefore, we obtain a system of $2M + 1$ differential equations that can be written, for each strain $i$, as follows:

$$\frac{dB_p^i}{dt} = (1-\lambda)G_p^i(R)B_p^i + \gamma \sum_{j=1}^{M} B_p^j B_0^i - dB_p^i, \qquad (7)$$

$$\frac{dB_0^i}{dt} = G_0^i(R)B_0^i + \lambda G_p^i(R)B_p^i - \gamma \sum_{j=1}^{M} B_p^j B_0^i - dB_0^i. \qquad (8)$$

Furthermore, if $\hat{R}$ represents the input of resource into the system, then

$$\frac{dR}{dt} = -\sum_{i=1}^{M}(u_p^i(R) + u_0^i(R)) - d(R - \hat{R}). \qquad (9)$$

Initial bacterial densities were determined by first running the system forward (with all strains initially present at equal densities) for $T = 24$ time units, and then clearing all plasmid-free cells from the population. This assumption is consistent with patients receiving antimicrobial therapy that clears all susceptible (plasmid-free) cells from the microbiota or, in an experimental microcosm, a round of growth in selective media after an overnight culture. As we are interested in the long-term population dynamics, we ran each simulation starting from the aforementioned initial condition until the plasmid fraction was below a threshold $\epsilon > 0$ (i.e., plasmid extinction), the plasmid fraction was near 100% and the total plasmid-free density was below $\epsilon$ (i.e., plasmid fixation), or wild-type and transconjugant subpopulations appeared to co-exist indefinitely in the population

(either in equilibrium or exhibiting oscillatory behaviour, as illustrated in Supplementary Figs. 10 and 11).

**Statistical analyses**. Analyses were performed using R (v. 3.5.0). Pearson's product-moment correlation was used to assess the linear relationship between growth curves parameters and relative fitness values obtained from competition assays, and between the relative fitness values obtained in the competition assays using isogenic competitors and those using *E. coli* J53/pBGC. To test the normal distribution of data from growth curves and competition assays, Shapiro–Wilk normality tests were performed. For normally distributed data, two-way ANOVA test and Tukey-Kramer honest significant difference were used to assess the reduction in relative fitness associated with the presence of the plasmid, and the differences in relative fitness values between species. Two-sampled *t* test with Bonferroni correction was used to assess the significance of pOXA-48_K8-associated fitness effects in the different bacterial hosts. Wilcoxon signed-rank test was used to compare the distribution of fitness effects obtained in our study with those form Vogwill and MacLean meta-analysis[46] (not normally distributed: Shapiro–Wilk normality test, $P = 0.0006$). Wilcoxon signed-rank test was also used to determine differences between species in relative $OD_{max}$, $\mu_{max}$, and AUC extracted from growth curves (Shapiro–Wilk normality test, $P < 0.004$ for all parameters except $\mu_{max}$ in *Klebsiella* spp., where $P = 0.215$). To determine differences in growth curve parameters between pOXA-48-carrying and pOXA-48-free isolates, paired Wilcoxon signed-rank tests with continuity correction were performed for *E. coli* isolates and *Klebsiella* spp. isolates.

**Reporting summary**. Further information on research design is available in the Nature Research Reporting Summary linked to this article.

## Data availability
The sequences generated and analysed during the current study and the annotated genomes of the isolates under study are available in the Sequence Read Archive (SRA), BioProject ID: PRJNA641166. Source data are provided with this paper.

## Code availability
The code generated during the current study is available in GitHub[56,86].

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

## Acknowledgements

This work was supported by the European Research Council under the European Union's Horizon 2020 research and innovation programme (ERC grant agreement no. 757440-PLASREVOLUTION) and by the Instituto de Salud Carlos III (co-funded by European Development Regional Fund "a way to achieve Europe") grant PI16-00860. R.C. acknowledges financial support from European Commission (grant R-GNOSIS-FP7-HEALTH-F3-2011-282512) and Plan Nacional de I+D+i2013–2016 and Instituto de Salud Carlos III, Subdirección General de Redes y Centros de Investigación Cooperativa, Ministerio de Economía, Industria y Competitividad, Spanish Network for Research in Infectious Diseases (REIPIR D16/0016/0011) co-financed by European Development Regional Fund "A way to achieve Europe" (ERDF), Operative programme Intelligent Growth 2014–2020. A.S.M. is supported by a Miguel Servet Fellowship (MS15-00012). J.R.-B. acknowledges financial support from Instituto de Salud Carlos III (Miguel Servet programme; CP20/00154), co-funded by ESF "Investing in your future" and Juan de la Cierva-Incorporación Fellowship (IJC2018-035146-I) co-funded by Agencia Estatal de Investigación del Ministerio de Ciencia e Innovación. M.H.-G. was supported with a contract from Instituto de Salud Carlos III, Spain (iP-FIS programme, ref. IFI14/00022). R.P.-M. was supported by PAPIIT-UNAM (IN209419) and CON-ACYT (Ciencia Básica grant A1-S-32164). We thank the Oxford Genomics Centre at the Wellcome Centre for Human Genetics (funded by Wellcome Trust grant reference 203141/Z/16/Z) for the generation and initial processing of the sequencing data.

## Author contributions

A.S.M., A.A.-d.V., and R.P.M. conceived the study. R.C. designed and supervised sampling and collection of R-GNOSIS bacterial isolates. M.H.-G. and P.R.-G. collected the bacterial isolates and performed bacterial characterisation. A.A.-d.V. performed the experimental work with help from J.R.-B. and J.D.F. A.A.-d.V., J.R.-B., and J.D.F. analysed experimental results. R.L.-S. and J.R.-B. performed the bioinformatic/phylogenetic analyses. R.P.M. developed the mathematical model and computer simulations. A.S.M. supervised the study. A.S.M., A.A.-d.V., and R.P.-M. wrote the initial draft of the manuscript, and all the authors contributed to the final version of the manuscript and approved it.

## Competing interests

The authors declare no competing interests.
