## [Peer Review File · Nature Communications]

REVIEWER COMMENTS

Reviewer #1 (Remarks to the Author):

In this manuscript Alonso-del Valle and colleagues studied the distribution of plasmid fitness effect in natural bacterial strains isolated from the human gut. A common perception in microbiology is that plasmid acquisition incurs a fitness cost to the host. Nonetheless, plasmids are common in nature (including the hospital environment, which is the topic of research in this manuscript). Several studies published in recent years (also in Nat Comm) aimed to reconcile plasmid ubiquity with their possible fitness effect on the host. Here, using a unique collection of natural isolates, the authors quantified the fitness effect of plasmid carriage in diverse strains of two species: *E. coli* and *K. pneumoniae*. The model plasmid in this study is a 'natural' conjugative plasmid of medium size (ca. 60Kb) and encodes for antibiotics resistance. Their results, overall, show that the plasmid has very often a negligible fitness effect on the host isolate. Additionally, the authors supplement their study with simulations of plasmid persistence in a mixed population depending on the plasmid fitness and transfer frequency. The results are clearly presented and the manuscript is well written. This study supplies a realistic view on the fitness effect (and hence evolutionary implications) of plasmid acquisition and as such, I expect that it will be interesting for Nat Comm readership and highly popular in the fields of microbial ecology and evolution (and likely also medical microbiology).

Note: I do not have the necessary background to evaluate the modelling approach. The results section of the paper is accessible and well explained. Still, I would try to connect it even better to the experimental part.

Comments

For using ANOVA one has to validate that the data fits the assumptions of that test. Specifically the test of normal distribution has to be performed and P value reported.

Figure 4b shows that only two of the tested isolates do not contain any plasmids. This should be mentioned in the text. I guess from the figure that there is no correlation between the fitness effect and the native plasmid content. Is that true?

The calibration of the fitness effect of pGBC is a neat solution (lines 499+). I'm missing some data here on the experimental design. Is the calculation performed with averaging or per replicate (i.e., with replicates also for the pGBC+/wt competition)?

The sequences are uploaded to NCBI, albeit, only as SRA, which makes them inaccessible for the common user. I would ask the authors to upload the assembled and annotated draft genomes to make them available to the community.

Suggestions

I see no reason to prefer the term 'quasi neutral' over neutral. I would replace the first with the latter.

The figures quality could improved – specifically the phylogenetic trees should be plotted with thicker branches and larger OTU font size.

Figure 2A – better plot proper boxplots (with raw data as dots) and separate the two species.

Figure 3 – could be better summarized with a cumulative distribution function (CDF), such that the choice of bin size has a smaller effect on the interpretation of the results.

Figure 4 – make it clear that the accessory gene tree is a gene content tree. Additionally, one could present the fitness data as color gradient on the tree branches.

For the correlation with the phylogenetic position – I would actually try to calculate a simple spearman correlation between the delta fitness and the pairwise distances that were used to construct the phylogenetic tree. A scatter plot of that data might be interesting (and revealing).

Figure 6 – would be nice to remind in the figure/legend what is sigma.

Reviewer #2 (Remarks to the Author):

This is an interesting study that expands our knowledge of plasmid-host interactions. The study composes two parts, first experimental measurement of the DFE of a plasmid across a clinical strain collection; second a model implementing these data into a population ecology model of plasmid dynamics.

The first part of the paper represents a lot of careful experimental work. The main result, that the DFE ranges from -20% to +20% fitness effects for a single plasmid is interesting. Moreover, these fitness effects seem to explain the distribution of the plasmid in the =clinic at least in Klebsiella. Given the plasmid came from Klebsiella originally, this stronger signal compared to E. coli is perhaps not unexpected. The work is clearly explained and thorough, with clear and informative figures and a simple but robust statistical analysis. I was at first concerned by the use of J53 in some of the competitions, but it is clear from the methods that this was carefully validated. A very nice job!

The model expands on the classical framework of Levin and colleagues. The main finding is that allowing for a distribution of fitness effects expands the parameter space in which plasmids survive. Moreover, plasmid stability scales positively with host diversity only when plasmid fitness effects can vary between hosts. The first result here is trivial and expected from Levin type models: in essence, if plasmids can sometimes be beneficial, then they can survive... The second result is more interesting and not immediately obvious from Levin style models.

The modelling is limited to the parameters derived from the experiments. This is fine in so far as it goes but does limit the conclusions to low cost plasmids. I think that the modelling shown in the paper should be expanded a bit to allow for a fuller exploration of parameter space e.g. I would expect that the effect of DFE is strongly dependent on the mean fitness effect as well as the variance, in that with larger costs the effect of the DFE diminishes because fewer plasmid-host interactions fall into the neutral/beneficial zone? For such high cost plasmids therefore, conjugation would be the main mechanism of maintenance

Overall I like the paper but feel that it is a bit oversold. The DFE perhaps "explains" persistence of low cost /nearly neutral plasmids, but is unlikely to explain it for other plasmid-host systems where the mean effect is a large cost and few/none of the associations fall into the neutral/beneficial FE space, for these persistence will still be "explained" by conjugation. I would like to see the title and some of the conclusions toned down a bit, and for an expanded set of conditions to be modelled, and appropriate caveats added to explain when DFE will matter most

Reviewer #3 (Remarks to the Author):

The manuscript by Alonso-del Valle et al. investigates the persistence of the antibiotic resistance plasmid pOXA-48 in an experimental setting using isolates of enterobacteria from the human gut. The authors examine the effect of pOXA-48 on the fitness of plasmid hosting strains. They find that while they often detect a fitness reduction in the host strains, in many isolates the plasmid does not have a negative impact on the host and may even be beneficial to the bacterial host. Using this data for a population model, the authors show that bacterial diversity may be more important than conjugation for the persistence of plasmids in natural bacterial communities. The study shows interesting aspects of plasmid persistence in a quasi-natural setting. I believe their results represent an important addition to the field of plasmid evolution and antibiotic resistance spread. Nevertheless, their study lacks certain depth and further investigation in the observed fitness effects of plasmids in natural clinical isolates is missing (especially in the genomics part). Furthermore, a major issue is the connection of the experimental part with the modelling part.

Major comments

1. The authors state in line 81-84 "Our criteria were to select (i) pOXA-48-free isolates, to avoid selecting

clones in which compensatory evolution had already reduced plasmid-associated costs; (ii) isolates from the most frequent pOXA-48-carrying species, *K. pneumoniae* and *Escherichia coli*; and (iii) strains isolated from patients located in wards in which pOXA-48-carrying enterobacteria were commonly reported". How can the authors be sure that the plasmid was not present in the strain before? According to their sampling scheme, it seems almost likely that the strains are commonly exposed to the plasmid. An analysis of the plasmid-carrying genomes versus plasmid-free genomes could help to answer that question. In any way, the authors cannot be sure that the plasmid never "saw" the host strain and the observed results may be very well explained by the evolutionary history of certain isolates.

2. In line 119-120 and Figure 2a statistics are missing and should be reported.

3. Is there a fitness effect of the plasmid pBGC carrying GFP in the natural isolates? GFP may have an effect on the fitness of the cells/culture. The authors should clarify this point in the main text as it may majorly influence the relative fitness calculation. Do the authors have a comparison of the fitness effect in the commonly used *E. coli* strain MG1655? Would be interesting to see the fitness effect in a potentially less closely related strain (for comparison).

4. Line 136: The results of the correlation analysis (statistics) should be reported in the main text: R values very low - should be considered & reported in the main text.

5. line 152, the authors report an ANOVA for all experiments. The authors should report the p-values of the fitness experiments as well. For example, the calculated p-values are missing in line 148-149.

6. I am not convinced of the relevance of comparing the results of the current study to another comparative study. The results of the Vogwill & maclean study are collected from very different experiments, especially laboratory settings. The authors should consider omitting this section.

7. Figure 4 shows very interesting results and represents an interesting approach to interpret the presented fitness data. The authors analyze the genomes of the host strains as well as their mobile elements in association with the observed fitness effects. They observe that some closely related isolates are negatively affected by the plasmid while others are positively affected. This finding fits to the observation of the plasmid spread shown in ref. 31. Such results could hint to compensatory evolution in some strains that often encounter the plasmid in a natural setting. The authors should clarify this point. The isolates that are negatively affected also have a 'high' plasmid load. Could some of these plasmids affect pOXA stability or cost? Furthermore, did the authors check for genetic signatures in plasmid-carrying strains that might be found also in plasmid-free strains (data from ref. 31)? This may help to further explain the differences in the plasmid fitness cost distribution.

8. Unfortunately, I am not an expert in mathematical models and cannot judge the conducted mathematical model in this study. Nonetheless, a major issue is the connection of the experimental part with the modeling part that is currently lacking. The authors need to justify the connection of the two

parts.

Minor comments

-The authors should consider to increase the font size in figures 1-4 (as it is very hard to read!).

-line 130: The authors state that they were unable to introduce pBGC into eight of the isolates and used E. coli strain J53 carrying the pBGC vector. The author need to mark (e.g., in figure 2b) which were these strains as it could have significant effect on the calculation of the relative fitness.

Reviewer #4 (Remarks to the Author):

This study first shows that the fitness effect of one particular, medically relevant plasmid varies greatly between strains of two medically relevant species known to carry this plasmid in clinical settings. The authors then assessed the effect of the variation in plasmid cost on plasmid persistence in bacterial communities whose members show such a diversity in plasmid fitness effects. The study questions are very significant and the results are important, especially of the beneficial effect of the plasmid on some strains, but there are also several points of concern, and the novelty is a bit overstated.

1. I am afraid the term “distribution of fitness effects (DFE)” is already being used extensively in evolution and population genetics with a slightly different meaning (DFE of new mutations). As the authors point out in their first sentence of the Discussion, “The DFE for new mutations is a central concept in genetics and evolutionary biology, with implications ranging from population adaptation rates to complex human diseases”. DFE refers however to the distribution of fitness effects of a mutation within a population of cells. This concept and its measurement have been at the center of key debates in evolutionary biology for at least 15 years and the authors are not citing much of this literature. I do not think DFE should be used in the context the authors are trying to use it – which is differences in fitness effects (of a plasmid) among different strains of either E. coli or K. pneumoniae that have multiple genetic differences and form a community (the term they use in the second part of their Results). This will only cause confusion in the literature, especially given the broad nature of this journal. The strains in the manuscript have multiple genetic differences that are not fully described, are too numerous to clearly determine how the strains are related to each other (mutational steps), and certainly not causally linked to the effect of the plasmid on host fitness. Even in future studies these differences are likely too numerous to allow for easy identification of which genomic differences (=genetic context) explain the heterogeneity in plasmid fitness cost.

The Distribution of Fitness effects involves the estimation of the total fitness effects of mutations as resulting from the sum of all the individual gene effects (i.e. the direct genotype to phenotype map) and the consideration of the so-called environmental effects, or the effects of the genetic context in which

the gene of interest is expressed. These cannot be distinguished here. The authors could use the terms heterogeneity or variability, but not this population genetics term DFE.

2. Novelty: This observation of variability in fitness effect of one plasmid in different hosts is very similar to what was first published by De Gelder et al. and Ponciano et al. back in 2007 (refs 23, 25), with the only difference that the main experimentally measured parameter was plasmid persistence (=stability), and plasmid fitness effects were measured (and estimated through modeling) for only a subset of the total number of strains. In principle it showed the same: that the fitness cost of a plasmid to its host – and therefore the long-term persistence - is very variable between hosts, even closely related strains. Moreover, a decade later Kottara et al (2018 - FEMS ME, 94 (1)) showed something very similar (from the abstract: “These data confirm that plasmid stability is dependent upon the specific genetic interaction of the plasmid and host chromosome rather than being a property of plasmids alone, and moreover imply that MGE dynamics in diverse natural communities are likely to be complex and driven by a subset of species capable of stably maintaining plasmids that would then act as hubs of HGT”). Therefore the following statements on L. 59-60 and L. 351 are not correct in my opinion: “In this study, we provide the first description of the distribution of fitness effects (DFE) of a plasmid in wild-type bacterial hosts”, and “but the DFE of a plasmid in multiple, ecologically compatible bacterial hosts had not been reported before”. The hosts used in the studies mentioned above were environmental strains, and in some of these studies several strains were from the same habitat. They all received the plasmid naturally by conjugation, just like in this study, and most were from an environment where that type of plasmid is frequently found, just like here for pOXA-48 .

If anything, this study shows that variability (i.e. stochastic population dynamics) is an essential component to better understand plasmid persistence, but demonstrating that concept in combination with experimental results is not new (see for instance abstracts in Ponciano et al. (2007) and De Gelder et al. (2007): “Also, we present a stochastic model in which the relative fitness of the plasmid-free cells was modeled as a random variable affected by an environmental process using a hidden Markov model (HMM). Extensive simulations showed that the estimates from the proposed model are nearly unbiased. Likelihood-ratio tests showed that the dynamics of plasmid persistence are strongly dependent on the host type. Accounting for stochasticity was necessary to explain four of seven time-series data sets, thus confirming that plasmid persistence needs to be understood as a stochastic process” and “Remarkably, a large variation in the stability of pB10 in different strains was found, even between strains within the same genus or species. ... “The findings of this study demonstrate that the ability of a so-called ‘BHR’ plasmid to persist in a bacterial population is influenced by strain-specific traits, ...”).

And then there is the study/studies by Hall et al (see comment 10).

All this takes away from the novelty of this study, even though it is going further than these older studies. It aims at assessing the effect of the variation in plasmid cost on plasmid persistence in bacterial communities whose members show a diversity in plasmid fitness effects. The study would be more novel if this concept would be the focus of this new study, and not the observation that there is variation between strains.

3. L. 49: “First, most experimental reports of fitness costs have studied arbitrary associations between plasmids and laboratory bacterial strains 7,24.” This relates to comment 2 above. I don’t agree with this statement. I don’t know why previously published plasmid-host associations would seem ‘arbitrary’. Many strains used to assess plasmid fitness cost and its amelioration over time have been rather recent environmental isolates (see also some of the papers from the Brockhurst group). They were relevant plasmid-host associations and ‘natural bacterial hosts’ (L. 51). Maybe there were ‘arbitrary’ lab strains in these two references 7 and 24 but not in all studies published so far.

4. The statistical estimation of the ODE model parameters via the Bayesian analysis is not validated, nor sufficiently explained. From the description of the methods and the results, it is not clear that the authors fully understand how to present their statistical analyses. It seems that the reaches and limitations of their analyses are not fully understood. MCMC is not, for instance, an inferential paradigm. The authors seem to confuse statistical models with an inferential approach (Bayesian or frequentist, for example). The authors should look at statistical/model oriented papers in ecology journals to see examples of how to fit population dynamics models. Furthermore, the authors do not provide evidence that the statistical method properly estimated the model parameters. The model is not checked, there is no comparison with the priors, and it is not shown how specifying the different priors affects the results. The authors should consider taking a simulating approach and simulate data sets of the exact same dimensionality as the observed data sets instead of simulating much longer data sets? Useful papers on how to validate Bayesian analyses are by A. Gelman and C.R. Shalizi (“Philosophy and the practice of Bayesian Statistics”) and Lele, S.R. (2020) (Consequences of lack of parameterization invariance of non-informative Bayesian analysis for wildlife management: Survival of San Joaquin kit fox and declines in amphibian populations. *Front. Ecol. Evol.* 7: 501. doi: 10.3389/fevo).

There are at least two other important concerns with the modeling portion of this manuscript:

- 1) When this paper is published and the data and code shared as stated in the last line of the manuscript, every analysis should be repeatable by any student or faculty interested in the topic. The diagnostic tests for their parameter estimation process should be transparent and made available as a supplement.
- 2) The authors should demonstrate unequivocally that the model parameters are uniquely identifiable and that the data they have is sufficient to tease apart the model parameter estimates. They should also demonstrate that their inferences are robust to model assumptions and repeatable under different prior parameterizations. The recent ecological literature has seen a flurry of papers presenting similar analyses but seldom practitioners stop to document the robustness of their inferences.

5. ‘Neutral’ (L. 213, 315, 374 ..) and ‘quasi-neutral’: (L. 23, 346, 355). While the data, and especially the variation in the data, supports the null hypothesis that there is no plasmid cost in several strains, I am not sure I would call these plasmids having a neutral effect (and quasi-neutral is not defined here). The method is simply not sensitive enough to detect very small fitness costs (or benefits) that may nonetheless have a long-term impact on the population dynamics of the organism. No detectable fitness effect is not the same as no fitness effect. That is, the absence of evidence is not evidence of absence.

6. Fig. S1 shows growth curves for transconjugants/plasmid-free strains, as does Fig 2. But where is the correction for the carriage of the plasmid with the fluorescent protein? That effect can differ between strains, yet it seems to be published separately: Fig. S10 shows the effect of that small plasmid separately, and there clearly is an effect in many strains. It is not sufficient to say “Note that the fitness effects of pBGC did not correlate with those from pOXA-48”. Even when there is not statistically significant correlation, for some strains the reported fitness effect of the pOXA plasmid may have been confounded by the presence of this plasmid, possibly leading to erroneous conclusions about how many strains were benefiting from or inhibited in their growth by the plasmid! The fitness effect of the pOXA plasmids should be corrected for the fitness effect of this small plasmid (in Figure 2, 3). From what I understand this was not done. (The authors have shown themselves in the past that there may be poorly understood interactions between co-residing plasmids, and these interactions may differ between host backgrounds).

7. L. 41-46: Another factor that tends to be forgotten: the fact that some plasmids have extremely high fidelity in their partitioning and on top of that the ability to inhibit the growth of plasmid-free segregants (psk). I honestly think that there is not such a strong paradox anymore when all these elements are considered.

8. It was not clear how the variability of the growth curve data (known to be quite noisy) from the 5 replicates per strain were considered in Fig. 1. Only one point per strain is shown but what is the variability within a strain? All the replicates should be plotted.

9. Title (“The distribution of plasmid fitness effects EXPLAINS plasmid persistence in bacterial communities”) AND L. 28 (“Our results provide a simple and general explanation for plasmid persistence in natural bacterial communities”). This sounds as the findings from this study are the ONLY explanation of plasmid persistence in bacterial communities. This should be toned down a bit in my opinion. “Diversity of plasmid fitness effects contributes to plasmid persistence in bacterial communities” would be more accurate as a title.

10. I missed discussion of the papers by James Hall and others (e.g. their source-sink paper in PNAS) that addressed the fate of a plasmid in a two-species community.

Minor Comments:

1. L. 21, L. 139, elsewhere: What are “ecologically compatible” strains?

2. L. 25 “set of conditions for plasmid stability in bacterial communities, with plasmid persistence increasing with bacterial diversity and becoming less dependent on conjugation”. This relates to the Results on L 311....: I understand that this implies that because some bacteria are better hosts than others, the more diversity of strains, the higher the probability that a plasmid will be retained in that

community. However, what if that strain (or strains) is/are a minority in the community? I think your approach assumes equal relative abundance for each type of host (=perfect evenness), but that is typically not the case ?

3. L. 177: The conclusion on Figure 3 (“Note that relative fitness values are normally distributed”) seems misleading to the reader: As the data for E coli and Klebsiella are stacked, the distribution looks indeed normal, but when looking at both species separately, the Klebsiella data do not seem as normally distributed as the E coli data. Is it OK to lump the data?

4. Fig. 4: It wasn't clear if the authors looked at correlation between fitness effect of the pOX plasmid and the families of other plasmids in the strains (inner circles). Are these data used at all in the analyses or interpretations? If not, why are they shown?

5. Fig. 4: I would recommend using another color scale than red-green, given there are many color-blind readers.

6. L. 307: “allowing fitness effects to vary between members of the population”. I think the authors mean ‘community’ here, as they defined above (L. 284: “To explore how plasmid stability is affected by increasing community complexity”

7. L. 376: “phylogeny might influence fitness compatibility between plasmids and bacteria at the clonal level,”. This should be reworded. I think the authors are trying to say that (as yet unknown) genetic differences between strains of the same species can explain differences in plasmid fitness cost. The word ‘phylogeny’ seems a bit too vague here. The next sentence is great though.

8. It would be helpful to some readers to indicate that pOXA-48 is an IncL/M- plasmid.

We would like to thank the reviewers for their helpful criticisms, which have allowed us to increase the quality of the manuscript. We have substantially revised the manuscript following the reviewers' suggestions, which we found particularly useful. The changes are highlighted in yellow in the revised manuscript.

We have introduced several significant changes that we briefly highlight here:

-We have tone down our statements and conclusions throughout the manuscript, and we have also modified the title. Moreover, we now highlight the effect of the variability of fitness effects in plasmid stability in complex polyclonal bacterial populations as the main novelty of the work.

-We have modified the figures to increase size and quality. We have included 5 new supplementary figures to address the reviewers' requests.

-We improved the connection between the experimental part and the mathematical model, and we have expanded the model to explore a broader parameter space.

-We have extended the genomic analysis to investigate the correlation between pOXA-48_K8 fitness effects and the native plasmid content of the wild type strains.

-We have extensively improved the controls of model parameterization and we include a file with the diagnostic plots of the MCMC algorithm for all strains used in this study (Supplementary File 1).

We provide a point-by-point response to the reviewers' comments here:

REVIEWER COMMENTS

Reviewer #1 (Remarks to the Author):

In this manuscript Alonso-del Valle and colleagues studied the distribution of plasmid fitness effect in natural bacterial strains isolated from the human gut. A common perception in microbiology is that plasmid acquisition incurs a fitness cost to the host. Nonetheless, plasmids are common in nature (including the hospital environment, which is the topic of research in this manuscript). Several studies published in recent years (also in Nat Comm) aimed to reconcile plasmid ubiquity with their possible fitness effect on the host. Here, using a unique collection of natural isolates, the authors quantified the fitness effect of plasmid carriage in diverse strains of two species: *E. coli* and *K. pneumoniae*. The model plasmid in this study is a 'natural' conjugative plasmid of medium size (ca. 60Kb) and encodes for antibiotics resistance. Their results, overall, show that the plasmid has very often a negligible fitness effect on the hosing isolate. Additionally, the authors supplement their study with simulations of plasmid persistence in a mixed population depending on the plasmid fitness and transfer frequency. The results are clearly presented and the manuscript is well written. This study supplies a realistic view on the fitness effect (and hence evolutionary implications) of plasmid acquisition and as such, I expect that it will be interesting for Nat Comm readership and highly popular in the fields of microbial ecology and evolution (and likely also medical microbiology).

We thank the reviewer for these comments.

Note: I do not have the necessary background to evaluate the modelling approach. The results section of the part is accessible and well explained. Still, I would try to connect it even better to the experimental part. In the new version of the manuscript, we included several modifications in Results sections to provide a better connection between the modelling section and the experimental part (lines 266-273, lines 297-304, lines 333-334).

Comments

For using ANOVA one has to validate that the data fits the assumptions of that test. Specifically the test of normal distribution has to be performed and P value reported.

We now include a detailed section of *Statistical analyses* in the Methods, where we explain the different tests performed, including the analysis used to test for normal distributions (lines 730-744). The results of these tests are presented in the Results section (lines 141-142, line 170, line 195, and line 202), or in the Methods section (lines 738-741).

Figure 4b shows that only two of the tested isolates do not contain any plasmids. This should be mentioned the text. I guess from the figure that there is no correlation between the fitness effect and the native plasmid content. Is that true?

We thank the reviewer for this insightful comment. We mention now that only 2 isolates do not carry any plasmids (lines 258-259). We have performed new analyses to investigate the potential correlation between the native plasmid content and the fitness effects of plasmid pOXA-48_K8. A simple correlation between the number of different plasmids in the host bacterium and pOXA-48 fitness effects produced no significant results:

Scatter plots illustrating the correlation analyses between the number of plasmids and the relative fitness of pOXA-48_K8-carrying isolates. An estimate of the Pearson correlation coefficient is provided within each plot, along with its p-value. The regression lines and confidence intervals at 0.95 (shadow) are represented.

However, we have performed a much more detailed analysis trying to link the plasmid profile (absence/presence of plasmids belonging to each plasmid family) with pOXA-48 fitness effects for *Klebsiella* spp. isolates, where the LIPA analysis revealed a significant association between the accessory genome and pOXA-48_K8 fitness effects for a subset of isolates. We performed a Factor Analysis of Mixed Data (FAMD),

which allows analysing datasets containing both quantitative and qualitative variables (new Supplementary Figure 6). Interestingly, this analysis revealed that the presence of plasmids belonging to the IncFIA or IncH1B families, and the absence of plasmids belonging to the IncFIB family, were associated with high pOXA-48_K8 costs in *Klebsiella* spp. isolates. We have included these results in the manuscript (lines 228-235), and in a new Supplementary Figure (Supplementary Figure 6).

The calibration of the fitness effect of pGBC is a neat solution (lines 499+). I'm missing some data here on the experimental design. Is the calculation performed with averaging or per replicate (i.e., with replicates also for the pGBC+/wt competition)?

We have tried to improve the explanation in the methods section. In the previous section we explained: "We used the collection of pBGC transformed wild type isolates as competitors against their isogenic pOXA-48-carrying and pOXA-48-free isolates. Specifically, two sets of competitions were performed for each isolate: pOXA-48-free vs. pBGC-carrying, and pOXA-48-carrying vs. pBGC-carrying. Five biological replicates of each competition were performed." (We specify now that the "pBGC-carrying" are the wild type pOXA-48-free isolates carrying pBGC, lines 540-541). We then included all the technical details about the competitions and followed with the method used for the calibration: "The fitness of the pOXA-48-carrying isolates relative to the pOXA-48-free parental isolates were calculated with the formula, $w_{pOXA-48+} / w_{pOXA-48-}$ to correct for the fitness effects of pBCG (see Supplementary Figure 13 for pBGC fitness effects), and the error propagation method was used to calculate the standard error of the resulting value." Therefore the calibration is performed using the average result of five independent biological replicates of the competition pOXA-48-carrying vs. pBGC-carrying ($w_{pOXA-48+}$), divided by the average result of five independent biological replicates of the competition pOXA-48-free vs. pBGC-carrying ($w_{pOXA-48-}$). To integrate the error of each average we used the error propagation method to calculate the standard error of the calibrated result. We have included a new sentence in the methods section to clarify this part (lines 561-564).

The sequences are uploaded to NCBI, albeit, only as SRA, which makes them inaccessible for the common user. I would ask the authors to upload the assembled and annotated draft genomes to make them available to the community.

We agree with the reviewer. We have now uploaded all the assembled and annotated draft genomes to the NCBI under the same accession code (BioProject ID PRJNA641166, <https://www.ncbi.nlm.nih.gov/sra/PRJNA641166>). We include this information in the methods section (lines 599-600), and in the data availability statement (lines 746-747). The BioSample Accession codes of these genomes are SAMN15344961 to SAMN15345007. We scheduled these sequences for immediate release, although it may take GenBank a few weeks to make the annotated genomes available and provide the final accession numbers.

Suggestions

I see no reason to prefer the term 'quasi neutral' over neutral. I would replace the first with the latter.

We thank the reviewer for this suggestion. We have removed quasi-neutral from the paper, and we have also removed neutral, as suggested by reviewer 4.

The figures quality could improved – specifically the phylogenetic trees should be plotted with thicker branches and larger OTU font size.

We have included the changes suggested by the reviewer.

Figure 2A – better plot proper boxplots (with raw data as dots) and separate the two species.

We have changed figure 2A and we now use proper boxplots instead of violin plots, and separate the two species.

Figure 3 – could be better summarized with a cumulative distribution function (CDF), such that the choice of bin size has a smaller effect on the interpretation of the results.

We agree with the reviewer that a CDF helps to interpret the results presented in Figure 3. Therefore, we have included insets in both panels in Figure 3 representing the CDF of the different datasets (Figure 3 and lines 195-197, and 202-204 in the figure legend).

Figure 4 – make it clear that the accessory gene tree is a gene content tree. Additionally, one could present the fitness data as color gradient on the tree branches.

We have clarified in the figure legend that the accessory genome tree is a gene content tree (line 255). We tried to present the fitness data as a colour gradient on the branches, but we finally decided to maintain the outer circle with the colour-code instead, because we think it is easier to interpret for the reader.

For the correlation with the phylogenetic position – I would actually try to calculate a simple spearman correlation between the delta fitness and the pairwise distances that were used to construct the phylogenetic tree. A scatter plot of that data might be interesting (and revealing).

We thank the reviewer for this suggestion. We have followed this advice and studied these correlations.

First, we performed a correlation comparing all the distances and delta fitnesses by for *E. coli* and *Klebsiella* spp.:

Scatter plots illustrating the correlation analyses between the delta fitness and the pairwise distance obtained from the phylogenetic trees and dendrograms of the core genome and the accessory genome respectively for *Escherichia coli* and *Klebsiella* spp. An estimate of the Pearson correlation coefficient is provided within each plot, along with its p-value. The regression lines and confidence intervals at 0.95 (shadow) are represented.

The results of these correlations are qualitatively similar to those we obtained with the *phyloSignal* R package¹, which include different tools specifically designed to identify statistical dependence between a given continuous trait (relative fitness) and the phylogenetic tree of the taxa from which the trait is measured. These methods are Abouheif's C_{mean} , Moran's I index, Bloomberg's K and K*, and Pagel's λ ¹. All methods except Pagel's λ detected a marginally significant phylogenetic signal in the *K. pneumoniae* core genome (Supplementary table 3 [first tab]; $0.11 > P > 0.02$). Abouheif's C_{mean} and Moran's I (but not Bloomberg's K and K*, and Pagel's λ) also detected a marginally significant signal in the *K. pneumoniae* accessory genome tree (Supplementary table 3 [first tab]; $P < 0.056$ for both cases). The main difference is that in this new correlation we observed a small signal for the core genome of *E. coli* isolates, but non-significant after all. Moreover, we also performed the same analysis for each different clone; the correlation between the delta fitness and phylogenetic distance compared to the remaining clones of the same genus:

Klebsiella spp. core

Klebsiella spp. accesory

E. coli core

E. coli accessory

Scatter plots illustrating the correlation analyses between the delta fitness and the pairwise distance obtained from the phylogenetic trees and dendrograms of the core genome and the accessory genome respectively for each isolate of *Escherichia coli* and *Klebsiella* spp. An estimate of the Pearson correlation coefficient is provided within each plot, along with its p-value. The regression lines and confidence intervals at 0.95 (shadow) are represented. The confidence intervals shadowed in red indicates a p-value < 0.05.

Interestingly, in these analyses we obtained similar results to those obtained with the Local Indicator of Phylogenetic Association (LIPA) analysis, which is probably not that surprising. Specifically, we found significant correlations both for accessory and core genomic content for *K. pneumoniae* ST1427 isolates (and also for the accessory genome in Kpn19, which we did not detect with LIPA).

In summary, we consider that our analysis using the *phylosignal* R package and LIPA are quite robust, because these tools are specifically designed to look for associations of traits with phylogeny. Therefore, we think that including these new correlation analyses in the new version of the manuscript would not necessary improve the study, although we are happy to include them as supplementary figures if the reviewer thinks this may help.

Figure 6 – would be nice to remind in the figure/legend what is sigma.

We have included this information in the figure legend (line 380).

Reviewer #2 (Remarks to the Author):

This is an interesting study that expands our knowledge of plasmid-host interactions. The study comprises two parts, first experimental measurement of the DFE of a plasmid across a clinical strain collection; second a model implementing these data into a population ecology model of plasmid dynamics.

The first part of the paper represents a lot of careful experimental work. The main result, that the DFE ranges from -20% to +20% fitness effects for a single plasmid is interesting. Moreover, these fitness effects seem to explain the distribution of the plasmid in the clinic at least in *Klebsiella*. Given the plasmid came from *Klebsiella* originally, this stronger signal compared to *E. coli* is perhaps not unexpected. The work is clearly explained and thorough, with clear and informative figures and a simple but robust statistical analysis. I was at first concerned by the use of J53 in some of the competitions, but it is clear from the methods that this was carefully validated. A very nice job!

We thank the reviewer for these comments.

The model expands on the classical framework of Levin and colleagues. The main finding is that allowing for a distribution of fitness effects expands the parameter space in which plasmids survive. Moreover, plasmid stability scales positively with host diversity only when plasmid fitness effects can vary between hosts. The first result here is trivial and expected from Levin type models: in essence, if plasmids can sometimes be beneficial, then they can survive... The second result is more interesting and not immediately obvious from Levin style models.

The modelling is limited to the parameters derived from the experiments. This is fine in so far as it goes but does limit the conclusions to low cost plasmids. I think that the modelling shown in the paper should be

expanded a bit to allow for a fuller exploration of parameter space e.g. I would expect that the effect of DFE is strongly dependent on the mean fitness effect as well as the variance, in that with larger costs the effect of the DFE diminishes because fewer plasmid-host interactions fall into the neutral/beneficial zone? For such high cost plasmids therefore, conjugation would be the main mechanism of maintenance.

We thank the reviewer for this constructive suggestion. We have followed the reviewer's advice and we have extended the parameter space analysed in our model to consider a wider range of average fitness effects. Specifically we have determined the conjugation threshold that positively selects for plasmids in the population as a function of the average fitness cost and assuming a range of variances of fitness effects (including zero variance). Interestingly, and as predicted by the reviewer, the effect of the heterogeneity in fitness effects is more important for plasmids producing a small cost, while plasmids producing a large cost depend more dramatically on a high conjugation rate. We have included this information in the manuscript (lines 297-304), and also as a new panel in Figure 5 (5d, lines 323-331).

Overall I like the paper but feel that it is a bit oversold. The DFE perhaps "explains" persistence of low cost /nearly neutral plasmids, but is unlikely to explain it for other plasmid-host systems where the mean effect is a large cost and few/none of the associations fall into the neutral/beneficial FE space, for these persistence will still be "explained" by conjugation. I would like to see the title and some of the conclusions toned down a bit, and for an expanded set of conditions to be modelled, and appropriate caveats added to explain when DFE will matter most.

We have toned down our statements and conclusions throughout the manuscript. We have also modified the title of the paper to tone down the message. We specified now that the variation in fitness effects would be most relevant for plasmids producing on average a small fitness reduction, while conjugation will still be the main factor promoting plasmid stabilization for those plasmids producing a large cost on average (lines 297-304 and lines 323-331). We also state this limitation in the Discussion section (lines 399-402). As we explain above we have also expanded the set of conditions modelled including different average fitness costs (Figure 5d).

Reviewer #3 (Remarks to the Author):

The manuscript by Alonso-del Valle et al. investigates the persistence of the antibiotic resistance plasmid pOXA-48 in an experimental setting using isolates of enterobacteria from the human gut. The authors examine the effect of pOXA-48 on the fitness of plasmid hosting strains. They find that while they often detect a fitness reduction in the host strains, in many isolates the plasmid does not have a negative impact on the host and may even be beneficial to the bacterial host. Using this data for a population model, the authors show that bacterial diversity may be more important than conjugation for the persistence of plasmids in natural bacterial communities. The study shows interesting aspects of plasmid persistence in a quasi-natural setting. I believe their results represent an important addition to the field of plasmid evolution and antibiotic resistance spread. Nevertheless, their study lacks certain depth and further investigation in the observed fitness effects of plasmids in natural clinical isolates is missing (especially in the genomics part). Furthermore, a major issue is the connection of the experimental part with the modelling part.

We thank the reviewer for the comments. In the new version of the manuscript, we provide a better connection between the experimental part and the modelling part (lines 266-273, lines 297-304, lines 333-

334). We also performed a deeper analysis connecting native plasmid content with pOXA-48_K8 fitness effects (see response to question #7 below).

Major comments

1. The authors state in line 81-84 “Our criteria were to select (i) pOXA-48-free isolates, to avoid selecting clones in which compensatory evolution had already reduced plasmid-associated costs; (ii) isolates from the most frequent pOXA-48-carrying species, *K. pneumoniae* and *Escherichia coli*; and (iii) strains isolated from patients located in wards in which pOXA-48-carrying enterobacteria were commonly reported”. How can the authors be sure that the plasmid was not present in the strain before? According to their sampling scheme, it seems almost likely that the strains are commonly exposed to the plasmid. An analysis of the plasmid-carrying genomes versus plasmid-free genomes could help to answer that question. In any way, the authors cannot be sure that the plasmid never “saw” the host strain and the observed results may be very well explained by the evolutionary history of certain isolates.

We agree with the reviewer. We cannot be 100% sure that the plasmid has not been present in the strain before. We selected pOXA-48-free isolates recovered from patients with no record of previous colonisation by pOXA-48-enterobacteria to reduce the chances of the previous pOXA-48 presence in the isolates (we have included this information in the new version of the manuscript, lines 81-82). Moreover, the proportion of patients colonized with pOXA-48-carrying enterobacteria during their hospital admission in the period when these isolates were recovered is low (1.13%, 105/9,275 patients)². However, it is impossible to rule out the possibility of previous pOXA-48 presence in these isolates. Therefore, we acknowledge this limitation in the discussion section of the new version of the manuscript (lines 433-437).

2. In line 119-120 and Figure 2a statistics are missing and should be reported.

We have included the results of the statistical analysis (lines 121-128).

3. Is there a fitness effect of the plasmid pBGC carrying GFP in the natural isolates? GFP may have an effect on the fitness of the cells/culture. The authors should clarify this point in the main text as it may majorly influence the relative fitness calculation. Do the authors have a comparison of the fitness effect in the commonly used *E. coli* strain MG1655? Would be interesting to see the fitness effect in a potentially less closely related strain (for comparison).

One important characteristic of pBGC is that GFP expression is inducible, so during the overnight competition GFP is repressed, we only induce its expression when we are about to measure the cell ratios in the flow cytometer. Therefore, the effect of GFP expression in the outcome of the competition is limited (this data is presented in the methods section, lines 133-134, lines 526-528, and we have added a new explanatory sentence in lines 547-548).

We measured the fitness effects of plasmid pBGC in the natural isolates, as presented in the previous version of the manuscript (Supplementary Figure 13). The fitness effects of pBGC are small in the majority of the isolates (average $w = 0.99$; $\text{Var} = 0.013$. Shapiro-Wilk normality test, $W = 0.89$, $P = 0.0006$), with most of the isolates presenting an alteration of relative fitness $< 5\%$. In fact, the fitness effect of pBGC across all the isolates is not significant (Wilcoxon signed rank exact test, $V = 547$, $p\text{-value} = 0.06$). Crucially, the way we performed the competitions the results would be robust even if pBGC produced a cost (we performed two independent competitions per isolate: pOXA-48-free vs. pBGC-carrying, and pOXA-48-carrying vs. pBGC-

carrying, and then extracted the relative fitness of pOXA-48-carrying compared to pOXA-48-free, explained in detail in lines 541-584). In *E. coli* J53³, a sodium azide resistant laboratory mutant of *E. coli* K-12 (MG1655 also derives from *E. coli* K-12), pOXA-48_K8 produces a large cost (approx. 20% reduction in relative fitness), while pBGC produced a 3.6% reduction of relative fitness in *E. coli* J53. However, *E. coli* K-12 (and by extension J53 and MG1655) belongs to the ST10, so it is related to the 6 wild-type *E. coli* ST10 isolates included in this work.

4. Line 136: The results of the correlation analysis (statistics) should be reported in the main text: R values very low - should be considered & reported in the main text.

We have included the R-values in the text and we specify that the correlations are weak (lines 144-146). In the new version of the manuscript we used two packages in R to calculate growth curve parameters: *growthrates* (which we already used in the previous version) and *flux* (which we used now to obtain the area under the growth curves [AUC], instead of the Gen5™ Microplate Reader software, line 510).

The fact that each independent growth curve parameter present a weak (but significant) correlation with the relative fitness values (*w*) is probably not that surprising, since *w* is affected by all of these parameters simultaneously, so it is maybe normal that none of them will explain all the variation in *w* independently. However, what we think is very significant is the fact that when we used the exact same growth curves data to model the population dynamics of each isolate (lines 274-282 and Supplementary File 1), and then perform *in silico* competition assays between plasmid-carrying and plasmid-free isolates (lines 283-290), the correlation between the relative fitness values obtained with the computational method and the ones obtained with the actual experimental competition assays is good ($R^2=0.6$, see Supplementary Figure 8), indicating that growth curves data and relative fitness data are strongly correlated.

5. line 152, the authors report an ANOVA for all experiments. The authors should report the p-values of the fitness experiments as well. For example, the calculated p-values are missing in line 148-149.

We have included this information in lines 160-162 "Two horizontal lines indicate those clones showing significant costs or benefits associated with carrying pOXA-48 plasmid (Bonferroni corrected two sample t-test, $P < 0.05$)", would the reviewer like us to present each independent p-value in a supplementary table?

6. I am not convinced of the relevance of comparing the results of the current study to another comparative study. The results of the Vogwill & Maclean study are collected from very different experiments, especially laboratory settings. The authors should consider omitting this section.

Our intention with this comparison is to highlight the importance of using "ecologically compatible", environmentally co-occurring, plasmids and bacteria when trying to understand the actual fitness effects produced by plasmids in their natural bacterial hosts. In this context, the comparison seems relevant to us since the works analysed in the meta-analyses include mostly combinations of plasmids and bacteria of very diverse origins (lines 183-184), in contrast to the ones in our study. Therefore, we would prefer to keep this section as part of the manuscript, because we believe that it helps to make the point that plasmid fitness effects are likely influenced by the ecological compatibility between plasmids and their bacterial hosts.

7. Figure 4 shows very interesting results and represents an interesting approach to interpret the presented fitness data. The authors analyze the genomes of the host strains as well as their mobile elements in

association with the observed fitness effects. They observe that some closely related isolates are negatively affected by the plasmid while others are positively affected. This finding fits to the observation of the plasmid spread shown in ref. 31. Such results could hint to compensatory evolution in some strains that often encounter the plasmid in a natural setting. The authors should clarify this point. The isolates that are negatively affected also have a 'high' plasmid load. Could some of these plasmids affect pOXA stability or cost?

We thank the reviewer for this suggestion. We have performed now a much more detailed analysis trying to link the plasmid profile (absence/presence of plasmids belonging to each plasmid family) with pOXA-48_K8 fitness effects for *Klebsiella* spp. isolates, where the LIPA analysis revealed a significant association between the accessory genome and pOXA-48_K8 fitness effects for a subset of isolates. We performed a Factor Analysis of Mixed Data (FAMD), which allows analysing datasets containing both quantitative and qualitative variables (new Supplementary Figure 6). Interestingly, this analysis revealed that the presence of plasmids belonging to the IncFIA or IncH1B families, and the absence of plasmids belonging to the IncFIB family, were associated with high pOXA-48_K8 costs in *Klebsiella* spp. isolates. However, since those specific plasmid profiles are closely associated with ST1427 clones in our collection, it is difficult to disentangle the role of core genome/accessory genome/plasmid content in determining the fitness costs of pOXA-48_K8 in this particular ST. Further work will be required to dissect the relative contribution of each of these genomic compartments. We have included these results in the manuscript (lines 228-235), and in a new supplementary figure (Supplementary Figure 6).

Moreover, we discuss the possibility of pre-existing compensatory adaptations affecting the magnitude of pOXA-48_K8 fitness effects in the isolates tested in the new version of the manuscript, as suggested by the reviewer (lines 433-437).

Furthermore, did the authors check for genetic signatures in plasmid-carrying strains that might be found also in plasmid-free strains (data from ref. 31)? This may help to further explain the differences in the plasmid fitness cost distribution.

This is a really interesting suggestion. This is an analysis that we are planning on doing in the near future, but if we want to do a comprehensive analysis we need to increase the sample size of plasmid-free isolates with sequenced genomes (and ideally measure pOXA-48_K8 fitness effects in those isolates too). We are planning on focusing on *Klebsiella pneumoniae* ST11, which is the most common host for pOXA-48-like plasmids in our hospital (and in many other hospitals in Spain/Europe). We have a large collection of pOXA-48-carrying ST11 from the hospital, but we need to increase the sample size of pOXA-48-free ST11 isolates (in this study we only included 4 *K. pneumoniae* ST11 isolates). We believe that using plasmid-free/plasmid-carrying isolates from the same ST will dramatically increase the power of a GWAS-type analysis. However, we believe that this particular analysis goes beyond the scope of the present paper.

8. Unfortunately, I am not an expert in mathematical models and cannot judge the conducted mathematical model in this study. Nonetheless, a major issue is the connection of the experimental part with the modeling part that is currently lacking. The authors need to justify the connection of the two parts.

In the new version of the manuscript, we included several modifications in Results sections to provide a better connection between the modelling section and the experimental part (lines 266-273, lines 297-304, lines 333-334).

Minor comments

-The authors should consider to increase the font size in figures 1-4 (as it is very hard to read!).

We have re-edited all the figures and increase size and quality. We also include the figures as independent files to facilitate their inspection.

-line 130: The authors state that they were unable to introduce pBGC into eight of the isolates and used *E. coli* strain J53 carrying the pBGC vector. The author need to mark (e.g., in figure 2b) which were these strains as it could have significant effect on the calculation of the relative fitness.

The names of these strains were indicated in the methods section on the competition assays (lines 566-567), but we have also included this information in the section of "Construction of pBGC, a GFP-expressing non-mobilizable plasmid" (line 534). We have indicated the competitions that were performed using *E. coli* J53/pBGC instead of the wild-type clone with pBGC in Figure 2b, as suggested. As we indicated in the methods section, we actually performed a control experiment to confirm that the results of the competitions performed against *E. coli* J53/pBGC showed a good correlation with those same competitions performed against the isogenic wild-type strain varying pBGC for a subset of 10 clones: "Results showed that relative fitness values calculated with isogenic competitions and those using *E. coli* J53/pBGC presented a good correlation (Pearson's correlation, $R= 0.81$, $t= 3.96$, $df= 8$, $P= 0.004$, Supplementary Figure 14)." (lines 577-579).

Reviewer #4 (Remarks to the Author):

This study first shows that the fitness effect of one particular, medically relevant plasmid varies greatly between strains of two medically relevant species known to carry this plasmid in clinical settings. The authors then assessed the effect of the variation in plasmid cost on plasmid persistence in bacterial communities whose members show such a diversity in plasmid fitness effects. The study questions are very significant and the results are important, especially of the beneficial effect of the plasmid on some strains, but there are also several points of concern, and the novelty is a bit overstated.

We thank the reviewer for the comments.

1. I am afraid the term "distribution of fitness effects (DFE)" is already being used extensively in evolution and population genetics with a slightly different meaning (DFE of new mutations). As the authors point out in their first sentence of the Discussion, "The DFE for new mutations is a central concept in genetics and evolutionary biology, with implications ranging from population adaptation rates to complex human diseases". DFE refers however to the distribution of fitness effects of a mutation within a population of cells. This concept and its measurement have been at the center of key debates in evolutionary biology for at least 15 years and the authors are not citing much of this literature. I do not think DFE should be used in the context the authors are trying to use it – which is differences in fitness effects (of a plasmid) among different strains of either *E. coli* or *K. pneumoniae* that have multiple genetic differences and form a community (the term they use in the second part of their Results). This will only cause confusion in the literature, especially

given the broad nature of this journal. The strains in the manuscript have multiple genetic differences that are not fully described, are too numerous to clearly determine how the strains are related to each other (mutational steps), and certainly not causally linked to the effect of the plasmid on host fitness. Even in future studies these differences are likely too numerous to allow for easy identification of which genomic differences (=genetic context) explain the heterogeneity in plasmid fitness cost.

The Distribution of Fitness effects involves the estimation of the total fitness effects of mutations as resulting from the sum of all the individual gene effects (i.e. the direct genotype to phenotype map) and the consideration of the so-called environmental effects, or the effects of the genetic context in which the gene of interest is expressed. These cannot be distinguished here. The authors could use the terms heterogeneity or variability, but not this population genetics term DFE.

We do not use the acronym “DFE” in the new version of the manuscript, as suggested by the reviewer, and we have reduced the use of “distribution of fitness effects”, replacing it by “variability of fitness effects” (also in the title), when possible. We do not use any parallelism with the DFE of spontaneous mutations anymore.

2. Novelty: This observation of variability in fitness effect of one plasmid in different hosts is very similar to what was first published by De Gelder et al. and Ponciano et al. back in 2007 (refs 23, 25), with the only difference that the main experimentally measured parameter was plasmid persistence (=stability), and plasmid fitness effects were measured (and estimated through modeling) for only a subset of the total number of strains. In principle it showed the same: that the fitness cost of a plasmid to its host – and therefore the long-term persistence - is very variable between hosts, even closely related strains. Moreover, a decade later Kottara et al (2018 - FEMS ME, 94 (1)) showed something very similar (from the abstract: “These data confirm that plasmid stability is dependent upon the specific genetic interaction of the plasmid and host chromosome rather than being a property of plasmids alone, and moreover imply that MGE dynamics in diverse natural communities are likely to be complex and driven by a subset of species capable of stably maintaining plasmids that would then act as hubs of HGT”).

Therefore the following statements on L. 59-60 and L. 351 are not correct in my opinion: “In this study, we provide the first description of the distribution of fitness effects (DFE) of a plasmid in wild-type bacterial hosts”, and “but the DFE of a plasmid in multiple, ecologically compatible bacterial hosts had not been reported before”. The hosts used in the studies mentioned above were environmental strains, and in some of these studies several strains were from the same habitat. They all received the plasmid naturally by conjugation, just like in this study, and most were from an environment where that type of plasmid is frequently found, just like here for pOXA-48.

If anything, this study shows that variability (i.e. stochastic population dynamics) is an essential component to better understand plasmid persistence, but demonstrating that concept in combination with experimental results is not new (see for instance abstracts in Ponciano et al. (2007) and De Gelder et al. (2007): “Also, we present a stochastic model in which the relative fitness of the plasmid-free cells was modeled as a random variable affected by an environmental process using a hidden Markov model (HMM). Extensive simulations showed that the estimates from the proposed model are nearly unbiased. Likelihood-ratio tests showed that the dynamics of plasmid persistence are strongly dependent on the host type. Accounting for stochasticity was necessary to explain four of seven time-series data sets, thus confirming that plasmid persistence needs to be understood as a stochastic process” and “Remarkably, a large variation in the stability of pB10 in

different strains was

found, even between strains within the same genus or species. ... “The findings of this study demonstrate that the ability of a so-called ‘BHR’ plasmid to persist in a bacterial population is influenced by strain-specific traits, ...”.

And then there is the study/studies by Hall et al (see comment 10).

All this takes away from the novelty of this study, even though it is going further than these older studies. It aims at assessing the effect of the variation in plasmid cost on plasmid persistence in bacterial communities whose members show a diversity in plasmid fitness effects. The study would be more novel if this concept would be the focus of this new study, and not the observation that there is variation between strains.

We thank the reviewer for this comment. We agree that in the previous version of the manuscript we probably failed to stress that the main novelty of this work is arguably not the variability of plasmid fitness effects in natural bacterial hosts, but the consequences of this variability on plasmid persistence in complex bacterial communities. We now highlight this concept as the main novelty (line 28, lines 396-406). Moreover, we have modified the title/abstract/discussion and all the statements indicated by the reviewer, and included Kottara *et al.* reference.

However, we do not agree with the reviewer regarding the lack of novelty of the experimental results: (i) the scale (50 wild-type isolates tested), (ii) the experimental approach (performing competition experiments with GFP-labelled wild type strains is extremely challenging and, because it's a direct measure of relative fitness, it provides more accurate results than indirect estimations of plasmid fitness effects), (iii) the phenotypic/genotypic comparisons (none of the works mentioned by the reviewer sequenced the genomes of the host bacterial strains and analysed at the distribution of fitness effects across phylogeny), (iv) and the clinical relevance of the clones/plasmid under study (a major carbapenem resistance conjugative plasmid and 50 isolates recovered from the gut microbiota of hospitalised patients) are completely unprecedented.

3. L. 49: “First, most experimental reports of fitness costs have studied arbitrary associations between plasmids and laboratory bacterial strains 7,24.” This relates to comment 2 above. I don't agree with this statement. I don't know why previously published plasmid-host associations would seem ‘arbitrary’. Many strains used to assess plasmid fitness cost and its amelioration over time have been rather recent environmental isolates (see also some of the papers from the Brockhurst group). They were relevant plasmid-host associations and ‘natural bacterial hosts’ (L. 51). Maybe there were ‘arbitrary’ lab strains in these two references 7 and 24 but not in all studies published so far.

We agree with the reviewer that “arbitrary” was not the best word to use in this context. We have changed this statement. We now explain most experimental reports of fitness costs have studied associations between plasmids and bacterial strains from different ecological origins (lines 48-50). We think that this claim is true, and we provide many references now to support it⁴⁻¹³. For example the R1 plasmid was isolated from *Salmonella paratyphi* and the experiments that look at costs and compensatory evolution of R1 used *E. coli* as hosts^{8,9} (employing classic strains used in labs such as MG1655 or J53). Or my own previous works, where we used *Pseudomonas aeruginosa* PAO1 type strain and antibiotic resistance plasmids from multiple different origins, or *E. coli* MG1655 and a lab engineered ColE1-type plasmid⁴⁻⁷. In fact it is very difficult to find examples studying plasmids and bacterial recipients from the same ecological origin, with the remarkable exception of works from the Brockhurst lab¹⁴⁻¹⁶, as the reviewer points out (we also highlight this fact in the text now, line 50).

4. The statistical estimation of the ODE model parameters via the Bayesian analysis is not validated, nor sufficiently explained. From the description of the methods and the results, it is not clear that the authors fully understand how to present their statistical analyses. It seems that the reaches and limitations of their analyses are not fully understood. MCMC is not, for instance, an inferential paradigm. The authors seem to confuse statistical models with an inferential approach (Bayesian or frequentist, for example). The authors should look at statistical/model oriented papers in ecology journals to see examples of how to fit population dynamics models. Furthermore, the authors do not provide evidence that the statistical method properly estimated the model parameters. The model is not checked, there is no comparison with the priors, and it is not shown how specifying the different priors affects the results. The authors should consider taking a simulating approach and simulate data sets of the exact same dimensionality as the observed data sets instead of simulating much longer data sets? Useful papers on how to validate Bayesian analyses are by A. Gelman and C.R. Shalizi ("Philosophy and the practice of Bayesian Statistics") and Lele, S.R. (2020) (Consequences of lack of parameterization invariance of non-informative Bayesian analysis for wildlife management: Survival of San Joaquin kit fox and declines in amphibian populations. *Front. Ecol. Evol.* 7: 501. doi: 10.3389/fevo).

We agree with the reviewer that MCMC is not an inferential paradigm. Our intention when using MCMC is to calibrate parameters of the population dynamics model to evaluate the stability of plasmids in synthetic communities composed of strains with a distribution of plasmid fitness effects analogous to the observed experimentally. As we are not interested in assigning biological interpretation to the estimated growth kinetic parameters, we have removed from the new version of the manuscript any statement that could be interpreted as we are using the MCMC algorithm to make inferences about the metabolic properties of the cell. We have also included a section in Methods detailing the Metropolis-Hastings MCMC algorithm used to parametrize our model (lines 675-695). We have also included a Supplementary File with diagnostic plots showing that chains have good mixing, as well as the maximum likelihood estimates for different prior distributions (Supplementary File 1). We also performed Gelman-Rubin convergence diagnostics tests, obtaining values close to one for all strains.

There are at least two other important concerns with the modeling portion of this manuscript:

1) When this paper is published and the data and code shared as stated in the last line of the manuscript, every analysis should be repeatable by any student or faculty interested in the topic. The diagnostic tests for their parameter estimation process should be transparent and made available as a supplement.

We agree with the reviewer on the importance of repeatability and transparency in science. We have included new figures (Supplementary File 1) illustrating diagnostic tests performed after using the MCMC algorithm to fit growth kinetic data of each strain in our collection. Code and data used to parametrize the model and produce the figures can be downloaded from <http://www.github.com/esb-lab/pOXA48/> (line 750-751).

2) The authors should demonstrate unequivocally that the model parameters are uniquely identifiable and that the data they have is sufficient to tease apart the model parameter estimates. They should also demonstrate that their inferences are robust to model assumptions and repeatable under different prior

parameterizations. The recent ecological literature has seen a flurry of papers presenting similar analyses but seldom practitioners stop to document the robustness of their inferences.

We thank the reviewer for this comment. The estimability and robustness of the MCMC algorithm used to parametrize a simple Monod model to OD data was established in a previous study⁵. In any case, we have included in the new version of this manuscript the results of a data-cloning method used to assess parameter identifiability^{17,18}. Figures in Supplementary File 1 (panels f-g) show that, as the number of data clones increases, the marginal posterior distribution of both growth kinetic parameters, specific affinity (V_{max}/K_m) and cell efficiency (ρ), converges to a multivariate normal distribution with a mean equal to the maximum likelihood estimate, thus suggesting that both parameters are uniquely identifiable. To assess the robustness of the maximum likelihood estimates when considering different priors, we repeated the parametrization of all strains for uniform, beta, gamma and lognormal distributions, obtaining similar posterior distributions, as illustrated in Supplementary File 1 (panels c-d). An ANOVA test allowed us to reject the null hypothesis that there are significant differences in estimates when considering different priors (p -value >0.05), therefore concluding that the MCMC algorithm is repeatable under different prior parametrizations¹⁹.

5. 'Neutral' (L. 213, 315, 374 ..) and 'quasi-neutral': (L. 23, 346, 355). While the data, and especially the variation in the data, supports the null hypothesis that there is no plasmid cost in several strains, I am not sure I would call these plasmids having a neutral effect (and quasi-neutral is not defined here). The method is simply not sensitive enough to detect very small fitness costs (or benefits) that may nonetheless have a long-term impact on the population dynamics of the organism. No detectable fitness effect is not the same as no fitness effect. That is, the absence of evidence is not evidence of absence.

We thank the reviewer for this comment. We do not use either "neutral" or "quasi-neutral" in the new version of the manuscript. Instead, we explain that the plasmid produces moderate effects in most of the bacterial hosts tested (lines 122, 302).

6. Fig. S1 shows growth curves for transconjugants/plasmid-free strains, as does Fig 2. But where is the correction for the carriage of the plasmid with the fluorescent protein?

In growth curves, the bacterial isolates are cultivated as pure cultures (only one clone per growth curve, not two). The growth curve data was obtained from pure cultures of pOXA-48_K8-free and pOXA-48_K8-carrying wild type isolates. Therefore, there is no correction to be done here, because none of those isolates express GFP. On the other hand, in the competition assays two different clones compete in the same culture. We used the pBGC-carrying (GFP-expressing once induced with arabinose) wild type isolates for the competition assays, and in this case the data was corrected to remove the effect of pBGC, as we explained in the methods sections (lines 535-584).

That effect can differ between strains, yet it seems to be published separately: Fig. S10 shows the effect of that small plasmid separately, and there clearly is an effect in many strains. It is not sufficient to say "Note that the fitness effects of pBGC did not correlate with those from pOXA-48".

Again, the fitness effects of pBGC are accounted for, and the relative fitness is calculated removing the effect of pBGC (see lines 558-564). Please see also response to the third comment from Reviewer #1 and to comment number 3 from Reviewer #3.

Even when there is not statistically significant correlation, for some strains the reported fitness effect of the pOXA plasmid may have been confounded by the presence of this plasmid, possibly leading to erroneous conclusions about how many strains were benefiting from or inhibited in their growth by the plasmid! The fitness effect of the pOXA plasmids should be corrected for the fitness effect of this small plasmid (in Figure 2, 3). From what I understand this was not done. (The authors have shown themselves in the past that there may be poorly understood interactions between co-residing plasmids, and these interactions may differ between host backgrounds).

This correction had been done and is explained in detail in the methods section.

7. L. 41-46: Another factor that tends to be forgotten: the fact that some plasmids have extremely high fidelity in their partitioning and on top of that the ability to inhibit the growth of plasmid-free segregants (psk). I honestly think that there is not such a strong paradox anymore when all these elements are considered. We thank the reviewer for this comment. pOXA-48_K8 carries in fact a PSK system (PemI/PemK). However, as we explained in the methods section (lines 487-493), despite this system in 7 out of the 50 isolates we found a very small % of plasmid-free colonies after propagating cultures in LB with no antibiotic selection (two consecutive days, 1:10,000 dilution) and plating cultures on LB agar. Moreover, using CRISPR-Cas9 technology we are able to cure pOXA-48 from wild-type isolates (data from a different project, not published), and this would be impossible if the PSK were completely effective. Therefore, these systems are not 100% infallible. In the introduction of the paper we just reflect the state of the art in the field, which includes the “plasmid paradox”.

8. It was not clear how the variability of the growth curve data (known to be quite noisy) from the 5 replicates per strain were considered in Fig. 1. Only one point per strain is shown but what is the variability within a strain? All the replicates should be plotted.

In supplementary Figure 1 we represented the curves of each isolate, including the 95% confidence intervals. In the new version of the manuscript we also include now a new Supplementary Figure (2) with all the independent replicates for OD_{max}, AUC and μ_{\max} values, as suggested by the reviewer. In any case, the results from the growth curves are complemented with the results from the competition assays, which provide a direct measure of relative fitness.

9. Title (“The distribution of plasmid fitness effects EXPLAINS plasmid persistence in bacterial communities”) AND L. 28 (“Our results provide a simple and general explanation for plasmid persistence in natural bacterial communities”). This sounds as the findings from this study are the ONLY explanation of plasmid persistence in bacterial communities. This should be toned down a bit in my opinion. “Diversity of plasmid fitness effects contributes to plasmid persistence in bacterial communities” would be more accurate as a title.

We agree with the reviewer and we have changed the title following her/his suggestion to “Variability of plasmid fitness effects contributes to plasmid persistence in bacterial communities”. We have also changed the statement in line 28.

10. I missed discussion of the papers by James Hall and others (e.g. their source-sink paper in PNAS) that addressed the fate of a plasmid in a two-species community.

We completely agree with the reviewer in this point. In the new version of the manuscript we discuss JPJ Hall works (lines 403-406).

Minor Comments:

1. L. 21, L. 139, elsewhere: What are “ecologically compatible” strains?

We refer to strains being ecologically compatible with the plasmid because plasmid and are “Environmentally co-occurring” (they are both in the hospital and in the gut of patients). We included this explanation now (line 67).

2. L. 25 “set of conditions for plasmid stability in bacterial communities, with plasmid persistence increasing with bacterial diversity and becoming less dependent on conjugation”. This relates to the Results on L 311...: I understand that this implies that because some bacteria are better hosts than others, the more diversity of strains, the higher the probability that a plasmid will be retained in that community. However, what if that strain (or strains) is/are a minority in the community? I think your approach assumes equal relative abundance for each type of host (=perfect evenness), but that is typically not the case?

This is an interesting point. In our model we assume equal relative abundance of each strain in the community. As we state in the discussion this is a simple model that does not consider spatial structure, complex ecological interactions between community members, plasmid-host co-evolution, or differential rates of horizontal transmission, which is a limitation (lines 437-440). Further work will be needed to increase the complexity of the model, as we state in the discussion section, including different relative abundances of the potential plasmid hosts.

3. L. 177: The conclusion on Figure 3 (“Note that relative fitness values are normally distributed”) seems misleading to the reader: As the data for *E coli* and *Klebsiella* are stacked, the distribution looks indeed normal, but when looking at both species separately, the *Klebsiella* data do not seem as normally distributed as the *E coli* data. Is it OK to lump the data?

We have analysed the data for *E coli* and *Klebsiella* spp. separately, and they are both normally distributed, so we think it is ok to lump the data. We include this information in the new version of the manuscript (lines 141-142).

4. Fig. 4: It wasn't clear if the authors looked at correlation between fitness effect of the pOX plasmid and the families of other plasmids in the strains (inner circles). Are these data used at all in the analyses or interpretations? If not, why are they shown?

We have now included a new and detailed analysis of the correlation between fitness effect of the pOXA-48_K8 plasmid and the families of other plasmids in the strains (lines 228-235, and Supplementary Figure 6). See also the response to comment #7 by Reviewer #3.

5. Fig. 4: I would recommend using another color scale than red-green, given there are many color-blind readers.

We thank the reviewer for this comment. The red-green palette we use is actually colour-blind friendly. There is general agreement that worldwide 8% of men and 0.5% of women have a colour vision deficiency. The 8%

of colour-blind people can be divided approximately into 1% deuteranopes, 1% protanopes, 1% protanomalous and 5% deuteranomalous (according to <https://www.colourblindawareness.org/colour-blindness/types-of-colour-blindness/>). Using the “Colorblinding” extension in Google Chrome this is the result:

Not colour-blind:

Deuteranopes:

Protanopes

Protanomalous

Deuteranomalous

6. L. 307: “allowing fitness effects to vary between members of the population”. I think the authors mean ‘community’ here, as they defined above (L. 284: “To explore how plasmid stability is affected by increasing community complexity”

We use now community, as suggested by the reviewer (line 358).

7. L. 376: “phylogeny might influence fitness compatibility between plasmids and bacteria at the clonal level,”. This should be reworded. I think the authors are trying to say that (as yet unknown) genetic differences between strains of the same species can explain differences in plasmid fitness cost. The word ‘phylogeny’ seems a bit too vague here. The next sentence is great though.

We thank the reviewer for this comment. We have replaced our sentence with the one suggested by the reviewer (lines 423-424).

8. It would be helpful to some readers to indicate that pOXA-48 is an IncL/M- plasmid.

Agreed, we have included this information in line 68.

References

1. Keck, F., Rimet, F., Bouchez, A. & Franc, A. Phylsignal: An R package to measure, test, and explore the phylogenetic signal. *Ecol. Evol.* **6**, 2774–2780 (2016).
2. Leon-Sampedro, R. *et al.* Dissemination routes of the carbapenem resistance plasmid pOXA-48 in a hospital setting. *bioRxiv* 2020.04.20.050476 (2020). doi:10.1101/2020.04.20.050476
3. Matsumura, Y., Peirano, G. & Pitout, J. D. D. Complete genome sequence of *Escherichia coli* J53, an azide-resistant laboratory strain used for conjugation experiments. *Genome Announc.* **6**, (2018).
4. San Millan, A., Heilbron, K. & MacLean, R. C. Positive epistasis between co-infecting plasmids promotes plasmid survival in bacterial populations. *ISME J.* **8**, 601–612 (2014).
5. San Millan, A. *et al.* Positive selection and compensatory adaptation interact to stabilize non-transmissible plasmids. *Nat. Commun.* **5**, 5208 (2014).
6. San Millan, A. *et al.* Integrative analysis of fitness and metabolic effects of plasmids in *Pseudomonas aeruginosa* PAO1. *ISME J.* **12**, 3014–3024 (2018).
7. San Millan, A., Escudero, J. A., Gifford, D. R., Mazel, D. & MacLean, R. C. Multicopy plasmids potentiate the evolution of antibiotic resistance in bacteria. *Nat. Ecol. Evol.* (2016). doi:10.1038/s41559-016-0010
8. Dahlberg, C. & Chao, L. Amelioration of the Cost of Conjugative Plasmid Carriage in *Escherichia coli* K12. *Genetics* **165**, 1641–1649 (2003).
9. Dionisio, F., Conceição, I. C., Marques, A. C. R., Fernandes, L. & Gordo, I. The evolution of a conjugative plasmid and its ability to increase bacterial fitness. *Biol. Lett* **1**, 250–252 (2005).
10. Loftie-Eaton, W. *et al.* Compensatory mutations improve general permissiveness to antibiotic resistance plasmids. *Nat. Ecol. Evol.* (2017). doi:10.1038/s41559-017-0243-2
11. Wein, T., Hülter, N. F., Mizrahi, I. & Dagan, T. Emergence of plasmid stability under non-selective

- conditions maintains antibiotic resistance. *Nat. Commun.* **10**, 1–13 (2019).
12. De Gelder, L., Ponciano, J. M., Joyce, P. & Top, E. M. Stability of a promiscuous plasmid in different hosts: No guarantee for a long-term relationship. *Microbiology* **153**, 452–463 (2007).
 13. Humphrey, B. *et al.* Fitness of *Escherichia coli* strains carrying expressed and partially silent IncN and IncP1 plasmids. *BMC Microbiol.* **12**, (2012).
 14. Hall, J. P. J. *et al.* Environmentally co-occurring mercury resistance plasmids are genetically and phenotypically diverse and confer variable context-dependent fitness effects. *Environ. Microbiol.* **17**, 5008–5022 (2015).
 15. Hall, J. P. J., Wood, A. J., Harrison, E. & Brockhurst, M. A. Source-sink plasmid transfer dynamics maintain gene mobility in soil bacterial communities. *Proc. Natl. Acad. Sci. U. S. A.* **113**, 8260–8265 (2016).
 16. Harrison, E., Guymier, D., Spiers, A. J., Paterson, S. & Brockhurst, M. A. Parallel Compensatory Evolution Stabilizes Plasmids across the Parasitism-Mutualism Continuum. *Curr. Biol.* **25**, 2034–2039 (2015).
 17. Lele, S. R., Nadeem, K. & Schmuland, B. Estimability and likelihood inference for Generalized Linear Mixed Models using data cloning. *J. Am. Stat. Assoc.* **105**, 1617–1625 (2010).
 18. Lele, S. R., Dennis, B. & Lutscher, F. Data cloning: Easy maximum likelihood estimation for complex ecological models using Bayesian Markov chain Monte Carlo methods. *Ecol. Lett.* **10**, 551–563 (2007).
 19. Campbell, D. & Lele, S. An ANOVA test for parameter estimability using data cloning with application to statistical inference for dynamic systems. *Comput. Stat. Data Anal.* **70**, 257–267 (2014).

REVIEWERS' COMMENTS

Reviewer #1 (Remarks to the Author):

I am satisfied with the authors response to my comments. The manuscripts read well and also the modelling part is not clearer to me. This is an important contribution to the field.

Reviewer #2 (Remarks to the Author):

The authors have responded adequately to all of my original criticisms. The extended modelling is most welcome. The results of the modelling do confirm, however, that the importance of variable fitness effects is restricted to low cost plasmids only. This inevitably diminishes the general significance of the findings, although the authors have appropriately modified the relevant statements to reflect this.

Reviewer #3 (Remarks to the Author):

The authors of the manuscript 'Variability of plasmid fitness effects contributes to plasmid persistence in bacterial communities' answered most of the comments satisfactory.

The only small concern I have is still the analysis shown in Figure 3 that comprises a comparison to other plasmid-induced fitness effect experiments. I am not convinced that the comparison is valid due to the nature of the diverse experiments. Furthermore, I think it takes the focus from the beauty of their experiments, because it is one of the rarer cases using natural isolates and not only lab strains.

Their study is overall an interesting and important addition to the field of antibiotic resistance and plasmid biology.

Reviewer #4 (Remarks to the Author):

The author did a very thorough job of responding to all the comments and adjusting the manuscript accordingly.

I have only one minor comment:

IN the newly added text, L. 298: " the conjugation threshold that positively selects for plasmids in the population,": this should probably be rephrased slightly because conjugation does not 'select' for plasmids, it helps the plasmid to persist in the population (I think that is what the authors mean).

We would like to thank the reviewers for their helpful criticisms, which have allowed us to increase the quality of the manuscript. We have substantially revised the manuscript following the reviewers' suggestions, which we found particularly useful. We provide a point-by-point response to the reviewers' comments here:

REVIEWER COMMENTS

Reviewer #1 (Remarks to the Author):

I am satisfied with the authors response to my comments. The manuscripts read well and also the modelling part is not clearer to me. This is an important contribution to the field.

We would like to thank the reviewer for these comments.

Reviewer #2 (Remarks to the Author):

The authors have responded adequately to all of my original criticisms. The extended modelling is most welcome. The results of the modelling do confirm, however, that the importance of variable fitness effects is restricted to low cost plasmids only. This inevitably diminishes the general significance of the findings, although the authors have appropriately modified the relevant statements to reflect this.

We thank the reviewer for the constructive comments.

Reviewer #3 (Remarks to the Author):

The authors of the manuscript 'Variability of plasmid fitness effects contributes to plasmid persistence in bacterial communities' answered most of the comments satisfactory.

The only small concern I have is still the analysis shown in Figure 3 that comprises a comparison to other plasmid-induced fitness effect experiments. I am not convinced that the comparison is valid due to the nature of the diverse experiments. Furthermore, I think it takes the focus from the beauty of their experiments, because it is one of the rarer cases using natural isolates and not only lab strains.

We thank the reviewer for this comment. It is true that in Figure 3 we are comparing multiple diverse experiments with our own results from this study, and this could be a bit problematic. However, we think that this figure really helps to illustrate the idea that plasmid fitness effects are likely influenced by the ecological compatibility between plasmids and their bacterial hosts (our associations are compatible while most of those from previous works are not), and therefore we would like to keep it in the manuscript.

Their study is overall an interesting and important addition to the field of antibiotic resistance and plasmid biology.

We thank the reviewer for this comment.

Reviewer #4 (Remarks to the Author):

The author did a very thorough job of responding to all the comments and adjusting the manuscript accordingly.

We would like to thank the reviewer for these words.

I have only one minor comment:

IN the newly added text, L. 298: " the conjugation threshold that positively selects for plasmids in the population,": this should probably be rephrased slightly because conjugation does not 'select' for plasmids, it helps the plasmid to persist in the population (I think that is what the authors mean).

We have rephrased the sentence as suggested by the reviewer (lines 326-327).